# Better than Your Teacher: LLM Agents that learn from Privileged AI Feedback

**Sanjiban Choudhury**[1,*], **Paloma Sodhi**[*]

## Abstract

While large language models (LLMs) show impressive decision-making abilities, current methods lack a mechanism for automatic self-improvement from errors during task execution. We propose LEAP, an iterative fine-tuning framework that continually improves LLM agents using feedback from AI expert teachers. Our key insight is to equip the expert teachers with a *privileged state* —information available during training but hidden at test time. This allows even weak experts to provide precise guidance, significantly improving the student agent's performance without access to privileged information at test time. We evaluate LEAP on multiple decision-making benchmarks, including text-based games (ALFWorld), web navigation (WebShop), and interactive coding (Intercode Bash). Our experiments show that LEAP (1) outperforms behavior cloning and ReAct baselines (2) enables weak student models (e.g., Llama3-8B) to exceed performance of strong teacher models (GPT-4o), and (3) allows weak models to self-improve using privileged versions of themselves. We provide a theoretical analysis showing that LEAP's success hinges on balancing privileged information with student's realizability, which we empirically validate. Our code is available at https://leap-llm.github.io.

## 1 Introduction

Large language models (LLM) show impressive decision-making abilities across domains (Yao et al., 2022b; Jimenez et al., 2023; Sodhi et al., 2024). However, they struggle to learn and recover from errors encountered at test time (Madaan et al., 2024; Shinn et al., 2023). Prompting alone is insufficient, as it requires manually specifying exceptions and examples (Khot et al., 2022), which is difficult to scale and leads to intractably long context lengths (Liu et al., 2024).

We address the problem of fine-tuning LLM agents to improve their decision-making using online interaction data. Imitation learning offers a sample-efficient way to train such models, where a teacher corrects the student agent's actions (Ross et al., 2011). However, finding an expert capable of both demonstrating the task and correcting mistakes from any state the student visits is extremely challenging (Chen et al., 2020). The expert must not only know how to complete the task but also how to recover from the student's errors, which is often more complex than simply performing the original task (Walsman et al., 2022). For example, in a task like "Heat the mug and put it in the cabinet," if the student places the wrong object in the microwave, the expert must first backtrack and remove the incorrect object before proceeding with the task.

***Our key insight is to equip the expert teacher with access to a privileged state*** — information useful for solving the task but only available at train time (Vapnik et al., 2015). This includes hidden states of a partially observable environment or intermediate subgoals. For a task "Heat the mug and put it in the cabinet", the privileged state reveals the mug's location and steps for heating it (Fig. 1). With this knowledge, the expert first clears the microwave and then teaches the student effective search strategies, like checking common mug locations. Over time, this enables the student to learn general strategies for solving the task and recovering from errors, ultimately outperforming the expert, even without access to the privileged state at test time.

We propose **L**earning from **E**xperts with **A**ccess to **P**rivilege (LEAP), an iterative learning algorithm that fine-tunes LLM agents using privileged AI expert feedback. As shown in Fig. 1, LEAP begins

---

[1]Cornell University, NY, USA. [*]Equal contribution. Correspondence to: Sanjiban <sanjibanc@cornell.edu>, Paloma <paloma.sodhi@gmail.com>

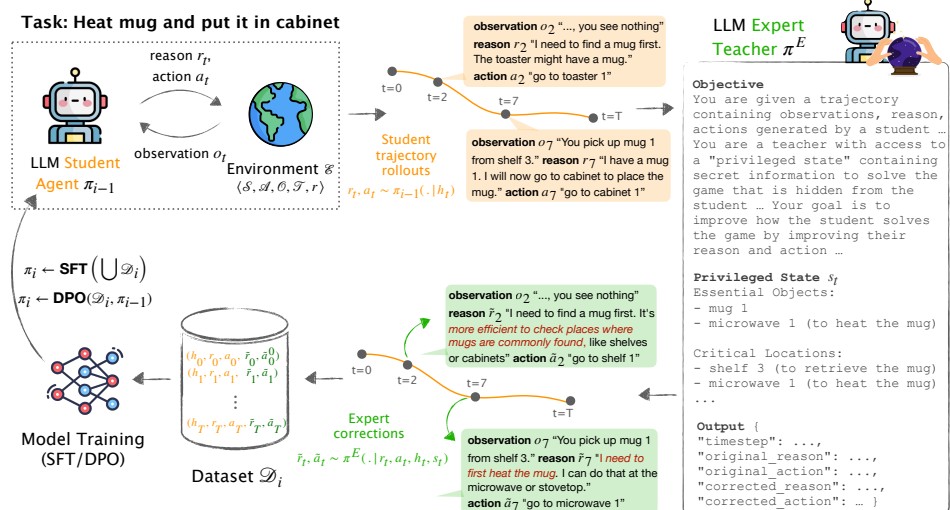

Figure 1: **LEAP overview**. LLM student agent interacts with the environment, generating a reason-action trajectory (in orange) based on its policy $\pi_{i-1}$. An expert teacher, with privileged state available only during training, evaluates and corrects the trajectory (in green). These corrections update the learner's policy to $\pi_i$ through SFT/DPO training. Updated policy $\pi_i$ is then rolled out at test time without access to privileged state.

with a student agent trained on demonstration data ($\pi_0$). During interactions, the student agent generates trajectories, and the privileged expert ($\pi^E$) identifies key time steps where corrections are needed. The expert provides improved reason and action labels based on the privileged state, which are then assembled into a dataset used to update the student model ($\pi_i \leftarrow \pi_{i-1}$). This process is repeated iteratively, progressively improving the student agent's performance.

Interestingly, we identify a key *trade-off between leveraging privileged information and the student agent's realizability*. While providing unrestricted privileged feedback allows for highly accurate corrections, these are often too complex for the student to act on effectively. Conversely, limiting feedback to what the student can easily process makes it realizable but often suboptimal. Our theoretical and empirical analyses show that balancing these extremes is key, which we achieve by introducing a constrained privileged expert that provides optimal yet realizable feedback.

Our key contributions are:

1. A novel iterative learning framework `LEAP` that fine-tunes LLM agents using privileged expert feedback, balancing privileged information and realizability.
2. Theoretical and empirical analysis of the optimal trade-off between privileged information and agent realizability.
3. Experimental validation on diverse interactive decision-making benchmarks: AlfWorld (Shridhar et al., 2020b), WebShop (Yao et al., 2022a), and InterCode (Yang et al., 2024).
   (a) `LEAP` consistently outperforms behavior cloning agents trained on `gpt-4o` or human demonstrations, showing significant improvements on ALFWorld ($65.7\% \rightarrow 91.8\%$), Web-Shop ($29.4 \rightarrow 61.8$), and InterCode Bash ($60.3\% \rightarrow 71.8\%$).
   (b) `LEAP` enables weaker models (e.g. `Llama3-8B`) to surpass stronger models (`gpt-4o`), with improvements on ALFWorld ($91.8\%$ vs. $65.7\%$), WebShop ($61.8$ vs. $58.4$), and InterCode Bash ($71.8\%$ vs. $71.3\%$).
   (c) Ablations that show LLM agents can self-improve using their privileged versions as teachers ($65.7\% \rightarrow 82.1\%$), explore the tradeoff between privileged information and realizability, and compare supervised finetuning with preference optimization.

## 2 PRELIMINARIES

**Problem Formulation: Decision-making LLMs.** We frame decision-making as a Partially Observable Markov Decision Process (POMDP) (Kaelbling et al., 1998) because real-world environments often contain hidden or incomplete information. The POMDP is defined by the tuple $\langle \mathcal{S}, \mathcal{A}, \mathcal{O}, \mathcal{T}, r \rangle$, where $\mathcal{S}$ represents the underlying state space of the environment, $\mathcal{A}$ the action space, $\mathcal{O}$ the observation space, $\mathcal{T} : \mathcal{S} \times \mathcal{A} \rightarrow \mathcal{S}$ the transition dynamics, and $r : \mathcal{S} \times \mathcal{A} \rightarrow \mathbb{R}$ the reward function. The

true state $s_t \in \mathcal{S}$ is not fully observable; instead, the agent receives partial observations $o_t \in \mathcal{O}$ at each timestep. Hence it must rely on the history $h_t \in \mathcal{H}$ of observation-actions to choose actions.

LLM-based agents are policies $\pi \in \Pi$, which predict both a *reason* $\rho_t$ and an action $a_t$ based on the agent's history $h_t$. The reason $\rho_t$ is expressed in natural language to justify action $a_t$. The history $h_t = o_1, a_1, \ldots, o_t$ is the sequence of past observations and actions. The policy is $(\rho_t, a_t) = \pi(h_t)$. While reinforcement learning (RL) for POMDPs is already challenging due to the large history space $\mathcal{H}$ (Papadimitriou & Tsitsiklis, 1987), this is further exacerbated by the addition of reason space, which is infinite and unconstrained. This makes RL impractical to run for LLM agents.

**Interactive Imitation Learning with DAGGER.** Imitation learning (IL) offers a promising alternative to RL, by allowing the agent to mimic an expert teacher policy $\pi^E$ bypassing the need for complex exploration. The gold-standard for imitation learning are interactive methods like DAGGER (Ross et al., 2011) that query the teacher $\pi^E$ for on-policy corrections on states $h_t$ visited by the student. However, for POMDPs, computing such corrections is intractable and strictly more challenging than just solving the original task, often resulting in suboptimal feedback.

## 3 APPROACH

We present **L**earning from **E**xperts with **A**ccess to **P**rivilege (LEAP), a framework for iteratively training an LLM agent through online AI feedback from a privileged expert (Fig. 1). Central to our approach is the concept of a *privileged expert teacher*, which leverages access to privileged state information to guide the student's recovery from errors. The goal is to first train the agent to solve the problem using this privileged state, but then distill the policy into a student that works from observations alone. Such frameworks have been used extensively in robotics for training robots in simulation and deploying in real world for applications like self-driving and legged locomotion (Sec. 5). We build on this paradigm, and propose novel extensions in the form of constrained privileged experts to adapt this for training LLM agents. Section 3.1 defines the privileged teacher, Section 3.2 details LEAP, and Section 3.3 presents the theoretical analysis of LEAP.

### 3.1 PRIVILEGED EXPERT TEACHER

We introduce a *privileged expert teacher*, a policy that has access to privileged state information $s_t$ which fully describes the environment at time $t$. This access transforms the expert's decision-making from a POMDP into a fully observable MDP making policy learning more tractable. Formally, the teacher policy takes the student's reason $\rho_t$, action $a_t$, history $h_t$, and privileged state $s_t$ and generates a corrected reason action $(\tilde{\rho}_t, \tilde{a}_t) \sim \pi^E(\cdot | \rho_t, a_t, h_t, s_t)$.

The privileged information $s_t$ can come from multiple sources:

1. *Simulator States:* Many agentic domains train in simulated environments before real-world deployment, where the simulator exposes underlying states. For instance, in AlfWorld (Shridhar et al., 2020a), agents observe only their location, but the simulator provides full item locations as privileged information during training.
2. *Evaluation Criteria:* Most domains have success metrics. In WebShop (Yao et al., 2022a), privileged information includes product attributes, options, and target prices, while in InterCode (Yang et al., 2024), it includes goal commands and unit tests.
3. *Human Annotations:* Privileged states can be extracted from human demonstrations. In AlfWorld, key subgoal details (e.g., "drawer 3 contains book 2") come from demonstrations. Similarly, dialog agents leverage conversation-level subgoal annotations, which are easier to collect than dense demonstrations, making them a practical alternative.

A crucial requirement for the privileged expert teacher is that *it must not reveal privileged information in its feedback to the learner*. The prompt instructions specify that the teacher should provide general reason and action that avoid the direct use of privileged information. Using such information directly might solve tasks quickly but makes the corrected actions $(\tilde{\rho}_t, \tilde{a}_t)$ unrealizable for the learner, who does not have access to the privileged state $s_t$, leading to a significant imitation gap and poor learner performance. In Section 3.3, we characterize the trade-off between using privileged information and student realizability. We show in Appendix B that a constrained privileged expert that effectively utilizes privileged information while remaining realizable for the student has the optimal performance. We approximate such an expert by designing prompts for the expert to instruct it to use privileged information, without revealing it. See Appendix E for prompts.

---

**Algorithm 1** `LEAP`: Iterative Learning with Privileged Expert Teacher

---

    **Input:** Privileged expert teacher $\pi^E$, Environment $\mathcal{E}$
    **Output:** Learned policy $\pi$
1: Collect an initial set of demonstrations from expert teacher $\mathcal{D} \leftarrow \{(h_t^*, \rho_t^*, a_t^*)\}$
2: Train BC $\pi_0$ on $\mathcal{D}$ using supervised fine-tuning (SFT)
3: **for** iteration $i = 1, 2, \ldots, N$ **do**
4:     Roll out policy $\pi_{i-1}$ in environment $\mathcal{E}$ to collect new data: $\mathcal{D}_i \leftarrow \{(h_t, \rho_t, a_t)\}$
5:     Compute privileged state $s_t$ for every datapoint in $\mathcal{D}_i$
6:     Invoke privileged expert to generate corrected reason and action: $\tilde{\rho}_t, \tilde{a}_t \sim \pi^E(.|\rho_t, a_t, h_t, s_t)$
7:     Augment dataset with corrected reason and action: $\mathcal{D}_i \leftarrow \{(h_t, \rho_t, a_t, \tilde{\rho}_t, \tilde{a}_t)\}$
8:     Train policy $\pi_i$ using a no-regret update: $\pi_i \leftarrow \text{SFT}\left(\bigcup \mathcal{D}_i\right)$ or $\pi_i \leftarrow \text{DPO}(\mathcal{D}_i, \pi_{i-1})$
    **return** Best $\pi \in \{\pi_1, \ldots, \pi_N\}$ on validation dataset

---

## 3.2 THE LEAP ALGORITHM

The `LEAP` algorithm iteratively trains the student policy $\pi$ through online interactions with the environment $\mathcal{E}$ and corrections from a privileged expert teacher $\pi^E$. The process begins by collecting an initial set of demonstrations of history, reason, and action from the teacher $\mathcal{D} \leftarrow \{h_t^*, \rho_t^*, a_t^*\}$, and training a BC policy $\pi_0$ on this data using supervised fine-tuning (SFT).

At each iteration, the current learner policy $\pi_i$ is rolled out in the environment to generate new data $\mathcal{D}_i \leftarrow (h_t, \rho_t, a_t)$. For each timestep in $\mathcal{D}_i$, `LEAP` computes the privileged state $s_t$ by leveraging information available only during training. The privileged teacher is then invoked on trajectories where the agent fails, generating corrected reasoning and actions, $\tilde{\rho}_t, \tilde{a}_t \sim \pi^E(.|\rho_t, a_t, h_t, s_t)$. These corrections are used to augment the dataset $\mathcal{D}_i$. The policy $\pi_i$ is subsequently updated using this augmented dataset through a no-regret learning method. This iterative process continues, refining the policy over multiple iterations until the best-performing policy is selected based on validation.

The update step in `LEAP` should be designed to ensure no-regret learning, a property that guarantees the policy's cumulative performance will converge to that of the best possible policy in hindsight. The most common update is to aggregate data $\bigcup \mathcal{D}_i$, and train $\pi_i$ using supervised fine-tuning (SFT) on the aggregated dataset, as done in DAGGER (Ross et al., 2011). This is equivalent to Follow the Regularized Leader (FTRL), a no-regret update step. An alternate update is to treat the problem as a preference optimization problem where the corrected reason action $(\tilde{\rho}_t, \tilde{a}_t)$ is preferred over the student reason action $(\rho_t, a_t)$. Preference optimizers like Direct Policy Optimization (DPO) (Rafailov et al., 2024) or Kahneman-Tversky Optimization (KTO) (Ethayarajh et al., 2024) optimize a preference loss while regularizing to the base policy, in this case the previous policy $\pi_{i-1}$. This is equivalent to an Online Mirror Descent (OMD) update, which is also a no-regret update. We use SFT by default but compare it against preference optimization as an ablation in the results.

## 3.3 ANALYSIS

We briefly touch on the theoretical guarantees of `LEAP`, and refer the reader to Appendix. B for a full exposition. We define the realizability of an expert policy and show how it affects `LEAP` performance.

**Definition 3.1** (Average Realizability Gap). The average realizability gap $\epsilon(\pi^E, T)$ between the privileged expert $\pi^E$ and the best policy $\pi^\star$ over time horizon $T$ is:

$$\epsilon(\pi^E, T) := \sup_\pi \frac{1}{T} \sum_{t=1}^{T} \mathbb{E}_{s_t, h_t \sim d_t^\pi} ||\pi^E(a_t|s_t) - \pi^\star(a_t|h_t))||_1 \tag{1}$$

where $||.||_1$ is the L1 distance, $d_t^\pi$ is the induced distribution over state and history.

**Theorem 3.2** (`LEAP` with Privileged Expert). *Running $N$ iterations of `LEAP` with privileged expert $\pi^E$ yields at least one policy $\pi$ such that the time-average performance $\frac{1}{T} J(\pi)$ is bounded as:*

$$\frac{1}{T} J(\pi) \geq \frac{1}{T} \underbrace{J(\pi^E)}_{performance} - H(\pi^E) \left( \underbrace{\epsilon(\pi^E, T)}_{realizability} + \gamma(N) \right) \tag{2}$$

*where $\pi^E$ is the privileged expert, $H(\pi^E)$ is the recoverability coefficient of $\pi^E$, $\epsilon(\pi^E, T)$ is the average realizability gap, and $\gamma(N)$ is the average regret of the* DAGGER *update.*

The equation shows a trade-off between performance of the privileged expert (first term) and the realizability of the privileged expert (second term), which we show how to optimize in Appendix. B.

## 4 EXPERIMENTS

### 4.1 OVERVIEW OF RESULTS

We evaluate LEAP across 3 decision-making domains: ALFWorld, a text-based game, WebShop, a web navigation environment, and Intercode, an interactive coding environment. Below is a summary:

1. **LEAP improves upon base behavior cloning (BC) policy:** LEAP significantly improves the base BC policy $\pi_0$ over successive fine-tuning iterations, showing gains on AlfWorld ($65.7\% \rightarrow 91.8\%$), WebShop ($29.4 \rightarrow 61.8$), and InterCode ($60.3\% \rightarrow 71.8\%$). See Tables 1, 2, Fig. 3.
2. **LEAP enables weaker student models to surpass stronger teacher models:** Over successive iterations, student models (e.g., Llama3-8B) outperform their stronger teachers (e.g., GPT-4o), despite starting with lower or comparable performance. For example, on ALFWorld, the student model ($\pi_3 : 91.8\%$) surpasses the teacher ($\pi^E : 65.7\%$), with similar trends on WebShop ($\pi_2 : 61.8$ vs. $\pi^E : 58.4$) and InterCode ($\pi_3 : 71.8\%$ vs. $\pi^E : 71.3\%$). See Tables 1, 2, Fig. 3.
3. **LEAP requires a balance between using privileged information and generating realizable corrections:** Experts using minimal privileged information ($\pi_1^E : 73.1\%$) generate more realizable actions, while those with excessive privileged feedback ($\pi_5^E : 5.22\%$) produce unrealizable corrections. $\pi_3^E$ strikes a perfect balance achieving optimal performance ($\pi_3^E : 91\%$). See Fig. 5.
4. **LEAP enables LLM agents to self-improve by using their own privileged versions as experts:** LLM agents can self-improve by using their privileged versions as teachers. For example, in ALFWorld, Llama3-8B bootstraps itself through iterations, improving from $65.7\%$ to $82.1\%$ and finally to $91.8\%$. See Fig. 6.
5. **LEAP can be trained using both SFT or preference optimization:** Both SFT and preference optimization improve over the base BC policy ($\pi_0$), with SFT delivering stronger gains ($68.6\% \rightarrow 96.4\%$) compared to DPO ($68.6\% \rightarrow 74.3\%$) and KTO ($68.6\% \rightarrow 87.1\%$). See Table 3.

### 4.2 DOMAIN 1: TEXT-BASED GAMES

**Setup.** ALFWorld (Shridhar et al., 2020b) is a text-based game where agents complete household tasks (e.g., "heat mug and put it in cabinet") by navigating and interacting via text commands. Success requires planning subgoals, tracking progress, and efficiently searching for objects. Each task has 30 timesteps, spanning six categories, with a training set of 3,257 games and two evaluation sets: 139 in-distribution and 134 out-of-distribution games. We compare against prior works and extend ReAct (Yao et al., 2022b) with stronger models (gpt-4o[1], Claude[2], Gemini[3]) and a general instruction prompt. Our approach fine-tunes Llama3-8B (Dubey et al., 2024) with LoRA (Hu et al., 2021), using gpt-4o as a privileged expert teacher for 4 iterations. Privileged information includes essential objects, their locations, and optimal actions. See Appendix D.2, E.1 for details and prompts.

**Results.** Table 1 shows that LEAP significantly outperforms all baselines, with the best policy achieving $91.8\%$ success rate on out-of-distribution tasks. Iteration 1 of LEAP has the biggest performance gain ($65.7\% \rightarrow 91.0\%$), **leading to a student policy $\pi_1$ that surpasses the teacher gpt-4o** with higher success rate ($91.8\% > 65.7\%$) and lower actions ($11.3 < 20.2$). Note that performance improvements are not monotonic over iterations of LEAP, ($65.7\% \rightarrow 91.0\% \rightarrow 83.6\% \rightarrow 91.8\%$), however, all policies are better than the BC policy $\pi_0$. The regression from $\pi_1$ to $\pi_2$ are mainly from HEAT and LOOK tasks, which gets corrected immediately in the next iteration $\pi_3$. Finally. we note that LEAP policies generalize well from in-distribution to out-of-distribution tasks with only a slight dip in performance for the final policy $\pi_3$ ($94.2\% \rightarrow 91.8\%$).

Fig. 2 (a) shows an example of LEAP training and testing. A typical failure mode of $\pi_0$ is that it searches for items inefficiently, e.g. looking at shelves or drawers or cabinets one by one until time runs out. The expert teacher leverages the privileged state of where objects are to show the student how to search efficiently without revealing the object location, e.g. it's more efficient to search desks where cellphones are likely to be than drawers sequentially one after another. Training on this correction data, $\pi_1$ *generalizes this reasoning to new test-time tasks*, e.g. putting a watch in a safe. Fig. 2 (b) shows that while $\pi_0$ inefficiently searches shelves, $\pi_1$ searches likely locations for the watch, first the sidetable and then the dresser to solve the task. The main performance gains of LEAP over

---

[1] https://platform.openai.com/docs/models    [2] https://docs.anthropic.com/en/docs/about-claude/models
[3] https://ai.google.dev/gemini-api/docs/models/gemini

| Method | All tasks | | PICK tasks | CLEAN tasks | HEAT tasks | COOL tasks | LOOK tasks | PICK 2 tasks |
|---|---|---|---|---|---|---|---|---|
| | %suc↑ | #act↓ | %suc↑ | %suc↑ | %suc↑ | %suc↑ | %suc↑ | %suc↑ |
| BUTLER [1] | 35.0 | - | 50.0 | 74.0 | 83.0 | 91.0 | 39.0 | 65.0 |
| ReAct few-shot [2] | 57.0 | - | 65.0 | 39.0 | 83.0 | 76.0 | 55.0 | 24.0 |
| Autogen gpt-3.5 [3] | 77.0 | - | - | - | - | - | - | - |
| ExpeL gpt-3.5 [4] | 59.0 | - | - | - | - | - | - | - |
| Reflexion gpt-3 [5] | 88.0 | - | 75.0 | 90.3 | 91.3 | 90.5 | 88.9 | 94.1 |
| AdaPlanner gpt-3 [6] | 91.7 | - | 100.0 | 96.7 | 95.6 | 100.0 | 100.0 | 47.0 |
| ReAct gpt-4o | 65.7 | 20.2 | 91.7 | 35.5 | 56.5 | 52.4 | 100.0 | 76.5 |
| ReAct gpt-4o-mini | 29.9 | 25.5 | 33.3 | 25.8 | 17.4 | 14.3 | 66.7 | 29.4 |
| ReAct claude-3.5-sonnet | 76.1 | 19.0 | 95.8 | 61.3 | 60.9 | 81.0 | 88.9 | 76.5 |
| ReAct claude-3.5-haiku | 16.4 | 27.2 | 33.3 | 9.7 | 8.7 | 9.5 | 38.9 | 0.0 |
| ReAct gemini-1.5-flash | 19.4 | 26.3 | 41.7 | 12.9 | 13.0 | 19.0 | 16.7 | 11.8 |
| LEAP Llama3-8B | 0.7 | 29.8 | 0.0 | 0.0 | 0.0 | 4.8 | 0.0 | 0.0 |
| LEAP Llama3-8B $\pi_0$ | 65.7 | 18.6 | 66.7 | 74.2 | 73.9 | 66.7 | 66.0 | 35.3 |
| LEAP Llama3-8B $\pi_1$ | 91.0 | 11.9 | 83.3 | 90.3 | 91.3 | 95.2 | 94.4 | 94.1 |
| LEAP Llama3-8B $\pi_2$ | 83.6 | 13.1 | 87.5 | 90.3 | 73.9 | 95.2 | 66.7 | 82.4 |
| LEAP Llama3-8B $\pi_3$ | 91.8 | 11.3 | 87.5 | 93.5 | 91.3 | 90.5 | 94.4 | 94.1 |
| **Method** | **All tasks** | | **PICK tasks** | **CLEAN tasks** | **HEAT tasks** | **COOL tasks** | **LOOK tasks** | **PICK 2 tasks** |
| | %suc↑ | #act↓ | %suc↑ | %suc↑ | %suc↑ | %suc↑ | %suc↑ | %suc↑ |
| BUTLER [1] | 40.0 | - | 69.0 | 67.0 | 88.0 | 76.0 | 69.0 | 54.0 |
| ReAct gpt-4o | 54.3 | 22.9 | 91.4 | 33.3 | 31.2 | 12.0 | 84.6 | 66.7 |
| ReAct gpt-4o-mini | 40.0 | 26.1 | 11.1 | 12.5 | 4.0 | 53.8 | 20.8 | 20.8 |
| LEAP Llama3-8B | 2.9 | 29.5 | 8.6 | 0.0 | 0.0 | 0.0 | 7.7 | 0.0 |
| LEAP Llama3-8B $\pi_0$ | 68.6 | 16.9 | 88.6 | 51.9 | 56.2 | 80.0 | 76.9 | 50.0 |
| LEAP Llama3-8B $\pi_1$ | 96.4 | 10.5 | 100.0 | 100.0 | 100.0 | 92.0 | 92.3 | 91.7 |
| LEAP Llama3-8B $\pi_2$ | 90.0 | 11.4 | 94.3 | 92.6 | 87.5 | 88.0 | 92.3 | 83.3 |
| LEAP Llama3-8B $\pi_3$ | 94.2 | 10.3 | 100.0 | 96.3 | 100.0 | 87.5 | 100.0 | 83.3 |

Table 1: **AlfWorld Evaluation. (top)** 136 out-of-distribution games and **(bottom)** 140 in-distribution (max 30 actions). Baseline comparisons include [1] BUTLER (Shridhar et al., 2020a), [2] ReAct few-shot (Yao et al., 2022b), [3] Autogen (Shridhar et al., 2020a), [4] ExpeL (Zhao et al., 2024). Note [5] Reflexion (Shinn et al., 2023) and [6] AdaPlanner (Sun et al., 2024) make multiple attempts on the same test task, while we do not. We also add our own REACT instruction prompt with different models. LEAP with an 8B model across iterations ($\pi_1, \pi_2, \pi_3$) outperforms the stronger teacher ReAct gpt-4o.

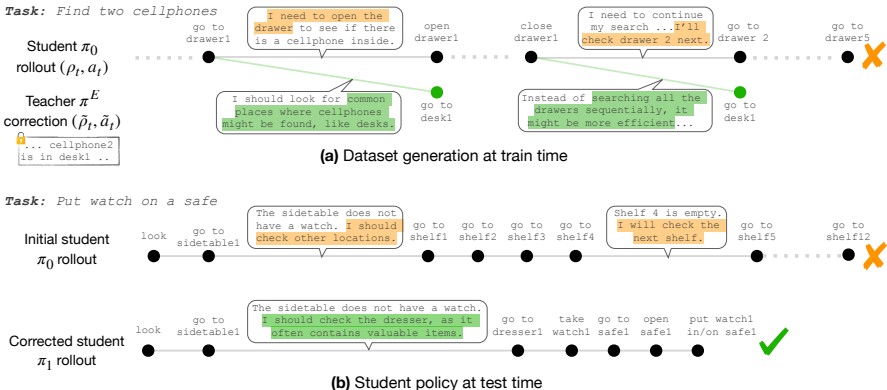

Figure 2: **ALFWorld Training and Testing for LEAP. (a) Training:** Student policy $\pi_0$ rolled out on training task to generate reason and actions, e.g. it fails to cell phone because it inefficiently searches all drawers. Expert teacher $\pi^E$ uses privileged information (cellphone in desk1) to generate general corrected reason actions that don't reveal privileged information (cellphones commonly found in desks, more efficient to search desks than drawers). **(b) Testing:** $\pi_0$ fails to find a watch as it inefficiently explores shelves one by one. $\pi_1$ learns a more efficient exploration policy, prioritizing areas like sidetables and dressers, solving the task quickly.

gpt-4o come from such efficient exploration. gpt-4o searches likely locations, however, its prior is from internet data and isn't necessarily aligned with the statistics of the ALFWorld environment. In contrast, LEAP leverages privileged feedback to learn this. See Appendix D for more results.

## 4.3 DOMAIN 2: WEB AGENTS

**Setup.** WebShop (Yao et al., 2022a) is an online shopping environment where agents fulfill user requests via search and click. Performance is based on attribute, option, product type, and price matches, with partial credit for partial matches. Each task has 30 timesteps, with 12,086 training and 500 test tasks. We compare against ReAct (Yao et al., 2022b) (GPT-4o, GPT-4o-mini) and the IL baseline (Yao et al., 2022a), which trains separate search and click models. Our approach fine-tunes

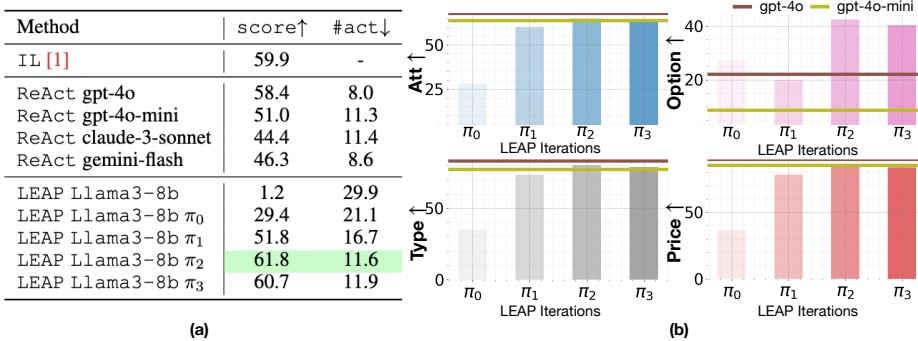

| Method | score↑ | #act↓ |
|---|---|---|
| IL [1] | 59.9 | - |
| ReAct gpt-4o | 58.4 | 8.0 |
| ReAct gpt-4o-mini | 51.0 | 11.3 |
| ReAct claude-3-sonnet | 44.4 | 11.4 |
| ReAct gemini-flash | 46.3 | 8.6 |
| LEAP Llama3-8b | 1.2 | 29.9 |
| LEAP Llama3-8b $\pi_0$ | 29.4 | 21.1 |
| LEAP Llama3-8b $\pi_1$ | 51.8 | 16.7 |
| LEAP Llama3-8b $\pi_2$ | 61.8 | 11.6 |
| LEAP Llama3-8b $\pi_3$ | 60.7 | 11.9 |

(a)

(b)

Figure 3: **WebShop Evaluation. (a)** Overall score↑ and #act↓ on 500 test tasks (max 30 actions). **(b)** Performance of LEAP over iterations on 4 different score components. Baseline comparisons include [1] IL (Yao et al., 2022a) and our ReAct instruction prompt with different models. LEAP with an 8B model across iterations ($\pi_2, \pi_3$) outperforms the stronger teacher ReAct gpt-4o.

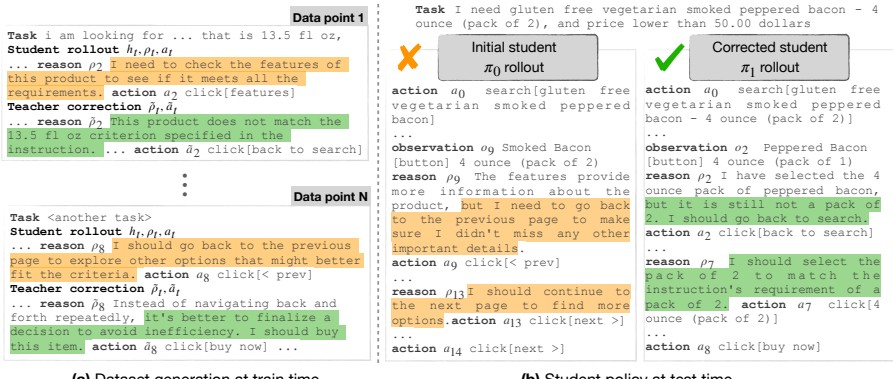

(a) Dataset generation at train time

(b) Student policy at test time

Figure 4: **WebShop Training and Testing for LEAP. (a) Training:** Teacher policy generates corrections on student rollout to backtrack when the product does not fit criteria or to commit to a product when it does. **(b) Evaluation:** $\pi_0$ fails to solve the task since it continues to search page after page even after discovering a good product. $\pi_1$ learns when to backtrack and when to commit to a product to solve the task in time.

Llama3-8B with LoRA, using gpt-4o as a privileged expert teacher. Privileged information includes product attributes, options, price constraints, and example product. See Appendix D.3, E.2 for details.

**Results.** Fig. 3 (a) shows that LEAP outperforms all baselines, with the best policy ($\pi_2$) achieving 61.4 score. Iteration 1 has the biggest performance gain ($29.4 \rightarrow 51.8$) followed by iteration 2 ($51.8 \rightarrow 61.8$). This **leads to a student policy $\pi_2$ that surpasses the teacher GPT-4o** with a higher score ($61.8 > 58.4$). Similar to ALFWorld, while performance improvements are not monotonic, both $\pi_2, \pi_3$ remain above the teacher gpt-4o performance. Fig. 3 (b) shows how LEAP improves on all 4 components of the score. The biggest improvements between LEAP and gpt-4o is in OPT component ($42.4 > 22.1$), indicating gpt-4o fails to select a product with the right option often.

Fig.4 (a) shows LEAP's correction of $\pi_0$ and $\pi_1$ using training data. The teacher leverages privileged information (e.g., product attributes or options) to guide the student in recognizing when to backtrack on mismatched items and when to commit to a suitable product. Training on this correction data produces policy $\pi_2$, which balances backtracking and committing effectively. Fig.4 (b) highlights $\pi_0$'s failure to commit, endlessly cycling through pages, whereas $\pi_2$ successfully backtracks and selects a suitable item to complete the task.

### 4.4 DOMAIN 3: INTERACTIVE CODING

**Setup.** Intercode Bash (Yang et al., 2024) is an interactive coding environment where agents solve filesystem tasks (e.g., "Find all text files and write their names to a single file") using Bash. Tasks are drawn from four NL2Bash (Lin et al., 2018) datasets; we train on the first two and test on the next two, with a 10-step time limit. We compare against ReAct (Yao et al., 2022b) using gpt-4o and gpt-4o-mini, as well as the top three entries on the Intercode leaderboard—TryAgain baselines

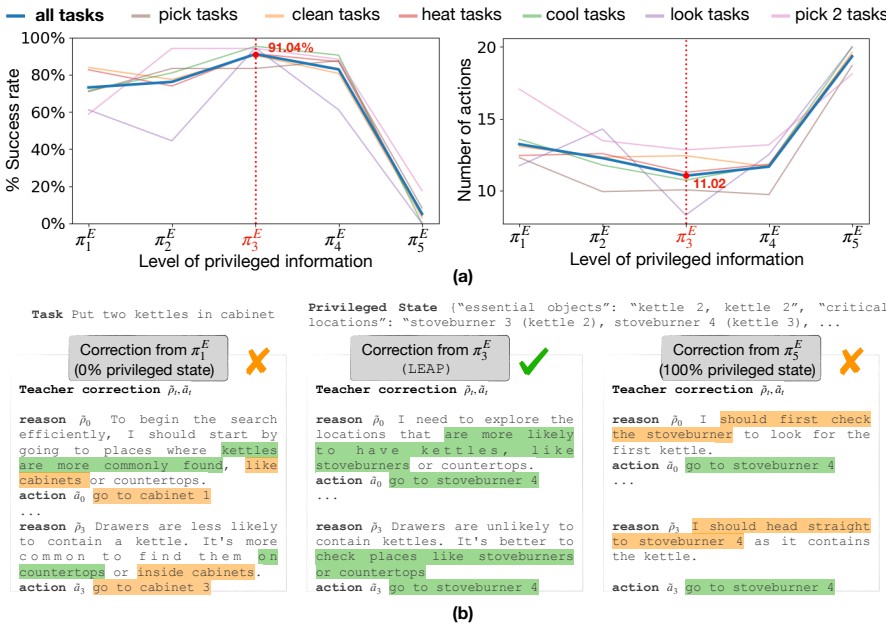

Figure 5: **Privileged Information vs Realizability. (a)** Performance of 5 policies trained with experts with varying levels of privileged information on ALFWorld, peaking for expert $\pi_3^E$. **(b)** Examples of corrections from experts $\pi_1^E, \pi_3^E, \pi_5^E$. $\pi_1^E$ generates realizable reason action but predicts wrong action. $\pi_5^E$ predicts correction action, but produces unrealizable reason action that contains privileged information. $\pi_3^E$ strikes a perfect balance.

(Yang et al., 2024) with `gpt-4`, `gpt-3.5`, and CodeLlama-34B-INST (Roziere et al., 2023). Our approach fine-tunes Llama-3.1-8B and Llama-3.1-70B with LoRA for two iterations, using `gpt-4o` as a privileged expert teacher. Privileged information consists of the ground truth Bash command.

**Results.** Table 2 shows that LEAP trained with 70B policy outperforms all baselines, with policy $\pi_1$ achieving 71.8% success rate to match the `gpt-4o` expert teacher (71.8% > 71.3%).

We find that on this coding benchmark, the `gpt-4o` expert is a very strong baseline. LEAP trained with 8B models improve over the BC policy, but falls short of the `gpt-4o` expert teacher. These results show that if the expert teacher is already a strong baseline, even after leveraging privileged information, it might be hard for the student to outperform the teacher.

| Method | %succ↑ | #act↓ |
|---|---|---|
| `gpt-4` [1] | 48.5* | - |
| `gpt-3.5` [1] | 46.5* | - |
| CodeLlama-34B-INST [1] | 36.0* | - |
| ReAct `gpt-4o` | 71.3 | 3.8 |
| ReAct `gpt-4o-mini` | 43.4 | 6.0 |
| LEAP Llama-3.1-8b $\pi_0$ | 41.4 | 6.6 |
| LEAP Llama-3.1-8b $\pi_1$ | 49.9 | 5.5 |
| LEAP Llama-3.1-70b $\pi_0$ | 60.3 | 4.8 |
| LEAP Llama-3.1-70b $\pi_1$ | 71.8 | 5.1 |

Table 2: **Intercode Bash Evaluation.** Overall success rate and actions on test data (max 10 actions). Baselines from [1] (Yang et al., 2024) evaluated on *(train + test) data.

### 4.5 WHAT IS THE TRADE-OFF BETWEEN PRIVILEGED INFORMATION AND REALIZABILITY?

We study the trade-off between how much privileged state the expert uses vs. the realizability of their corrections on ALFworld. We create 5 different privileged experts $\pi_E^1, \ldots, \pi_E^5$ by varying the prompt to instruct them to use increasing amounts of privileged state, e.g. for $\pi_E^2$ it says "Use this information sparingly for your correction," but for $\pi_E^4$ it says "feel free to include information from the privileged state". We run 1 iteration of LEAP to generate updated $\pi^1, \ldots, \pi^5$. Fig. 5 (a) shows that success rate initially rises from $\pi^1$ to $\pi^3$, but then falls off sharply till $\pi^5$. Num actions has a similar trend. This validates the hypothesis that there is indeed an optimal tradeoff. Interestingly, $\pi_5$ is much worse than $\pi_1$, showing that it's more important to be realizable than to provide optimal yet unrealizable corrections. Fig. 5 (b) shows the corrections from $\pi_1^E, \pi_3^E, \pi_5^E$. $\pi_1^E$ generates reasoning that is perfectly realizable to the student, but since it has no privileged state the actions are not optimal. In contrast, $\pi_5^E$ predicts the correct action, but blatantly reveals the privileged information in the reason, being unrealizable to the student. $\pi_3^E$ strikes a perfect balance where it offers the correct action, but offers a reason that is very much realizable, i.e. kettles are likely to be in stoveburners.

We run another experiment to examine the effect of the teacher providing on-policy corrections on the states visited by the student, detailed in the Appendix C.1. Our hypothesis is that LEAP works due to

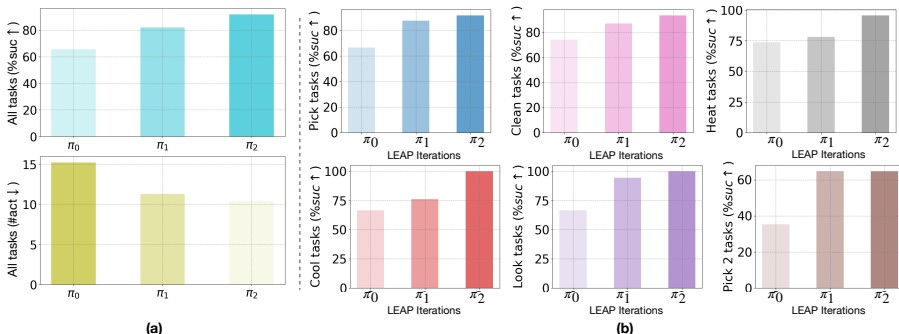

Figure 6: **Self Correction.** LEAP self-improves a Llama3-8B model that is used as both student and teacher. **(a)** Overall success rate and num actions. **(b)** Category-wise breakdown of success rate.

these two key factors: (a) the LLM teacher effectively balances the use of privileged information and (b) the LLM teacher provides on-policy corrections.

## 4.6    CAN LEAP BE USED TO SELF-CORRECT A STUDENT?

We test the hypothesis that LEAP should enable a model to self-improve by using its privileged version as the expert. On ALFWorld, we fine-tune Llama3-8B initializing with the BC policy $\pi_0$ and running 2 iterations of self-improvement. Fig. 6 shows that LEAP is able to significantly self-improve policies over iterations ($65.7\% \rightarrow 82.1\% \rightarrow 91.8\%$), with the final policy **matching the best policy when using GPT-4o as a teacher**. We see uniform improvements across all categories, with some improving from $\pi_0 \rightarrow \pi_1$ and others from $\pi_1 \rightarrow \pi_2$. We conclude that for some environments, like ALFworld, the privileged state does the heavy lifting, i.e. performance improvements happen at a similar rate to using a strong teacher vs. using the model itself as a teacher. We note that this need not hold for all environments, e.g. for some a strong teacher is required to extract the optimal reason and action from the privileged state. This effect is seen on WebShop in Appendix C.2 where while self-correction improves performance, it is unable to outperform the gpt-4o baseline.

## 4.7    HOW DOES SFT COMPARE TO PREFERENCE OPTIMIZATION?

We study the effect of different update methods for LEAP: SFT vs Preference Optimization. Our initial hypothesis is that KL regularized preference optimization should also result in performance improvement without needing to aggregate datasets. We run 1 iteration of DPO with different regularization $\beta$ values. Table 3 shows that while DPO improves upon the base $\pi_0$ policy, the improvement is far smaller than SFT for both $\beta = 0.1, 0.01$. One explanation is that SFT is far more aggressive in using the correction data compared to preference optimization, i.e., SFT deems the corrected reason action to be better than any alternative. To test this, we used a different preference optimizer KTO, that operates with unpaired preference data and set weight $\lambda_D = 1.0, \lambda_D = 0.0$, effectively emulating SFT-like behavior. KTO with these settings has a significantly high performance of $88.1\%$ close to SFT ($91.0\%$).

|  | Method | %succ↑ | #act↓ |
|---|---|---|---|
| in-dist. | LEAP $\pi_0$ | 68.6 | 16.9 |
|  | LEAP DPO ($\beta : 0.1$) $\pi_1$ | 74.3 | 15.7 |
|  | LEAP DPO ($\beta : 0.01$) $\pi_1$ | 74.3 | 15.6 |
|  | LEAP KTO ($\lambda_U : 0.0$) $\pi_1$ | 87.1 | 11.6 |
|  | LEAP SFT $\pi_1$ | 96.4 | 10.5 |
| out-dist. | LEAP $\pi_0$ | 65.7 | 18.6 |
|  | LEAP DPO ($\beta : 0.1$) $\pi_1$ | 71.6 | 17.9 |
|  | LEAP DPO ($\beta : 0.01$) $\pi_1$ | 70.9 | 17.7 |
|  | LEAP KTO ($\lambda_U : 0.0$) $\pi_1$ | 88.1 | 11.7 |
|  | LEAP SFT $\pi_1$ | 91.0 | 11.9 |

Table 3: **SFT vs Preference:** LEAP with SFT vs DPO vs KTO on ALFWorld.

## 5    RELATED WORK

**Imitation Learning and Privileged Information.** A powerful paradigm in machine learning is leveraging privileged information (Vapnik et al., 2015)—data available only during training and inaccessible at test time. This concept has been transformative in robotics, where a simulated expert policy with full state supervises a learner that only perceives sensor observations, such as navigation (Zhang et al., 2016; Uppal et al., 2024), self-driving (Chen et al., 2020), manipulation (Chen et al., 2023a; Hu et al., 2024), and legged locomotion (Lee et al., 2020; Kumar et al., 2021).

A fundamental challenge in utilizing privileged information is the realizability gap—where actions suggested by an expert may be infeasible for the learner to predict. This gap often leads to spurious correlations, manifesting as the "latching effect" where learners repetitively predict the same action,

commonly observed in autonomous driving (Muller et al., 2005; Kuefler et al., 2017; Bansal et al., 2018; Codevilla et al., 2019) and language models (Bengio et al., 2015; Holtzman et al., 2019; Wang & Sennrich, 2020; Ortega et al., 2021). While off-policy methods (Wen et al., 2020; 2022) have been proposed, (Swamy et al., 2022) show that online interaction with the environment is necessary.

Interactive imitation learning methods such as DAGGER (Ross et al., 2011), where experts provide corrective actions during the learner's rollouts, both theoretically and empirically lead to very effective policies in this regime (Choudhury et al., 2018). These methods operate under the assumption of asymptotic realizability (Swamy et al., 2022), where the learner's policy can eventually match the expert's actions as the episode progresses. However, when this assumption fails—often due to the partial observability of the task or model capacity—suboptimal performance ensues, particularly for LLM agents, as demonstrated by our experiments (Sec. 4.5). While recent approaches resort to exploration or RL (Walsman et al., 2022; Tennenholtz et al., 2021; Nguyen et al., 2022; Weihs et al., 2021), these are intractable to extend to LLM agents. Our work provides a practical recipe for LLM agents that optimally balances the trade-off between using privileged information and maintaining realizability, and further shows how such an expert can be used to enable policy self-improvement.

**Learning from AI Feedback.** A scalable method for aligning language model using AI feedback (Lee et al., 2023). Recent works have explored different types of feedback, from self-generated corrections to those provided by external AI models or environments (Pan et al., 2023). One body of work focuses on self-correction, where LLMs refine their responses based on feedback from their own outputs (Bai et al., 2022; Madaan et al., 2024). This can occur either at train-time, where LLMs are fine-tuned to generate corrected responses (Bai et al., 2022; Ganguli et al., 2023), or at test-time, where the model improves its response interactively (Madaan et al., 2024). However, while simple in principle, Huang et al. (2024) shows that without external sources of feedback, LLMs often struggle to accurately judge the correctness of their reasoning, limiting the effectiveness of purely self-generated corrections.

Another cluster of work explores feedback from external sources. Some approaches use feedback from strong models like GPT-4 acting as judges (Zheng et al., 2023) or critics trained on high-quality curated data (Wang et al., 2023; Paul et al., 2023). Similarly, An et al. (2024) and Guo et al. (2024) use GPT-4 to provide corrections and preference feedback. Welleck et al. (2023); Qu et al. (2024) trains LLMs to self-correct from errors. Others leverage environmental feedback or external tools to guide improvements. Chen et al. (2023b); Olausson et al. (2023) utilize execution feedback to improve code generation, and (Gou et al., 2023) employs external tools. Notably, Reflexion (Shinn et al., 2023) leverages feedback from external environments to generate self-reflections for agents, although it assumes multiple attempts in the same environment, which our approach does not. Our work introduces the use of privileged information in decision-making tasks for LLM agents, a technique not yet fully explored in the literature. This also enables LLM agents to self-improve using privileged information, setting it apart from existing self-improvement methods that rely on iterative solution generation and evaluation (Zelikman et al., 2022; Pang et al., 2024; Hosseini et al., 2024).

## 6 LIMITATIONS

We proposed LEAP, an iterative fine-tuning framework that improves LLM agent performance using privileged AI feedback, enabling weaker models to surpass stronger expert teachers and allowing agents to self-improve. However, LEAP has notable limitations. First, generating interaction rollouts is time-consuming, particularly in complex environments requiring multi-step reasoning. Each rollout requires simulating the environment and generating complete trajectories, that becomes computationally expensive across multiple fine-tuning iterations. Second, corrective feedback from the expert often results in lengthy output tokens. As trajectory lengths grow, balancing the effectiveness of feedback with its verbosity becomes challenging. Exploring methods to distill and summarize expert feedback at critical points is an interesting direction for future work.

## ACKNOWLEDGEMENTS

This work was supported in part by the NSF FRR (#2327973), NSF RI (#2312956), and ONR Young Investigator Award. Sanjiban Choudhury is supported in part by the Google Faculty Research Award and the OpenAI Superalignment Grant.

## REPRODUCIBILITY STATEMENT

To ensure the reproducibility of our findings, we have made several efforts, as outlined below. Detailed information and resources are provided in the main paper, appendix, and supplementary materials:

1. **Open source code.** We open source the code including the Python package implementing LEAP, along with all prompts, configuration files, and training scripts necessary to reproduce our results.
2. **Experimental details.** We provide descriptions of our experiments in the main paper (Section 4), with additional details on the setup, hyperparameters, prompts found in Appendix D, E. We use open-source datasets and models in our experiments, with citations and links.
3. **Analysis.** The key results of our theoretical analysis are presented in the main paper (Section 3.3), with full derivations, proofs, and assumptions detailed in Appendix B.

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

# Appendix

## Table of Contents

## A  BROADER IMPACTS

**Technological Impact.**    By iteratively fine-tuning LLMs through privileged expert feedback, LEAP can significantly enhance AI performance across diverse applications, such as virtual assistants and dialogue agents handling complex decision-making tasks. This approach allows LLMs to autonomously refine their decision-making, reducing the need for human intervention, streamlining automation, and improving efficiency in both consumer and enterprise environments.

Additionally, LEAP 's ability to enable weaker models to outperform stronger ones democratizes AI access, making state-of-the-art models more accessible to organizations with limited computational resources. This also opens new possibilities for improving model generalization in settings where perfect information is unavailable. However, the reliance on privileged feedback during training underscores the importance of further research into model transparency and interpretability. As these models grow more autonomous, ensuring their decision-making remains understandable to users will be critical for building trust and fostering widespread adoption.

**Societal Impact.**    Equipping LLMs with iterative self-improvement capabilities unlocks a range of societal benefits. In sectors such as education, customer service, and software development, these AI systems can assist professionals in decision-making, automate routine tasks, and enhance overall productivity. By continuously improving their performance, these models can reduce the cognitive load on workers, allowing them to focus on more complex, creative, and value-added activities.

Despite these advantages, there are ethical concerns that must be addressed. As LLMs become capable of self-improvement, ensuring their trustworthiness and accountability becomes critical. Without proper safeguards, the autonomous refinement of AI could lead to unpredictable behaviors, making it harder to ensure alignment with human values. Furthermore, there is a risk that self-improving AIs could be misused for malicious purposes, such as automating harmful tasks or spreading disinformation. To mitigate these risks, strict ethical guidelines and robust oversight is necessary to ensure deployment of such systems serves societal good and prevents misuse.

## B  ANALYSIS

### B.1  TRADE-OFF BETWEEN REALIZABILITY AND PRIVILEGED INFORMATION

We now analyze the performance of a policy imitating an expert with access to privileged information. The performance depends both on the performance of the expert policy and the realizability gap between the learner and the expert. We then derive a constrained privileged expert to better trade off these two terms and provide a practical approximation of this constrained expert. For notational simplicity, we assume that $a_t$ represents both reason and action. We follow the derivation in (Swamy et al., 2022) but simplify for our setting.

Let $\pi \in \Pi$ be the student policy that we are training. We assume the class of policies $\Pi$ is rich enough to contain all history-based policies. Let $J(\pi)$ be the performance, i.e. the cumulative reward of a learner policy that selects actions based on the history $a_t = \pi(h_t)$. Let $J(\pi_E)$ be the performance of the expert policy that selects actions based on the privileged state $a_t = \pi^E(s_t)$. Let $\tau \sim \pi$ be a *privileged rollout* which is the sequence of observation, actions, and privileged state upon executing the policy. We represent the rollout as $\tau = \{(h_t, s_t, a_t)\}$. Note that such a privileged rollout is only accessible at train time.

We begin by defining the average imitation gap between an expert policy $\pi^E$ and learner $\pi$.

**Definition B.1** (Average Imitaiton Gap)**.**  The average imitation gap of a learner $\pi$ over a horizon $T$:

$$\mathrm{AIG}(\pi, T) := \frac{1}{T} \left( J(\pi^E) - J(\pi) \right) = \mathbb{E}_{\tau \sim \pi^E} \left[ \frac{1}{T} r(s_t, a_t) \right] - \mathbb{E}_{\tau \sim \pi} \left[ \frac{1}{T} r(s_t, a_t) \right] \quad (3)$$

We next define the realizability gap between the privileged expert and the best policy in policy class as the L1 distance (or equivalently Total-Variation (TV) distance) between the two policies on the distribution of states and histories induced by *any policy*.

**Definition B.2** (Average Realizability Gap). The average realizability gap over time between the privileged expert $\pi^E$ and the best policy in the policy class $\pi^\star$ is:

$$\epsilon(\pi^E, T) := \sup_\pi \frac{1}{T} \sum_{t=1}^T \mathbb{E}_{s_t, h_t \sim d_t^\pi} ||\pi^E(a_t|s_t) - \pi^\star(a_t|h_t))||_1 \tag{4}$$

We next define the recoverability of the expert. Let $A^{\pi^E}(s, a) = Q^{\pi^E}(s, a) - V^{\pi^E}(s)$ be the advantage of a state action $(s, a)$ with respect to the expert policy $\pi^E$. A high negative advantage means that the mistaken action at a state results in a high value difference from the action the expert would have taken. Recoverability is the worst case advantage.

**Definition B.3** (Recoverability Coefficient of the Privileged Expert). The recoverability of the expert policy $\pi^E$ is the max advantage of the expert policy

$$H(\pi^E) := \max_{s,a} ||A^{\pi^E}(s, a)|| \tag{5}$$

We are now ready to state the main theorem of LEAP.

**Theorem B.4** (LEAP with Privileged Expert). *Running $N$ iterations of LEAP with privileged expert $\pi^E$ yields at least one policy $\pi$ such that*

$$\frac{1}{T} J(\pi) \geq \frac{1}{T} J(\pi^E) - H(\pi^E) \left( \epsilon(\pi^E, T) + \gamma(N) \right) \tag{6}$$

*where $\pi^E$ is the privileged expert, $H(\pi^E)$ is the recoverability coefficient of $\pi^E$, $\epsilon(\pi^E, T)$ is the average realizability gap, and $\gamma(N)$ is the average regret of the DAGGER update.*

The first term is the performance of the privileged expert and the second term is the average realizability of the privileged expert. The more privileged information the expert policy $\pi^E$ leverages the better $J(\pi^E)$ and the lower recoverability coefficient $H(\pi^E)$. However, the realizability gap $\epsilon(\pi^E, T)$ grows larger.

*Proof.* The proof relies on two important results.

The first is the Performance Difference Lemma (PDL) (Kakade & Langford, 2002) which states that the performance difference between any two policies can be expressed as the sum of advantages.

$$J(\pi) - J(\pi') = \sum_{t=1}^T \mathbb{E}_{s_t \sim d_t^\pi} \left[ \sum_{a_t} A^{\pi'}(s_t, a_t) \pi(a_t|s_t) \right] \tag{7}$$

where $s_t \sim d_t^\pi$ is the induced state distribution by $\pi$, and $A^{\pi'}(s_t, a_t) = Q^{\pi'}(s_t, a_t) - V^{\pi'}(s_t)$ is the advantage w.r.t. $\pi'$.

To adapt the equality to our setting where two policies operate on different input spaces, we note that the equality to different input spaces, e.g. state $s_t$ or history $h_t$ or joint state-history $s_t, h_t$. We express in terms of the joint state-history $s_t, h_t$ space and write the inequality as:

$$J(\pi) - J(\pi^E) = \sum_{t=1}^T \mathbb{E}_{s_t, h_t \sim d_t^\pi} \left[ \sum_{a_t} A^{\pi^E}(s_t, h_t, a_t) \pi(a|s_t, h_t) \right] \quad \text{(From (7))}$$

$$= \sum_{t=1}^T \mathbb{E}_{s_t, h_t \sim d_t^\pi} \left[ \sum_{a_t} A^{\pi^E}(s_t, h_t, a_t) \pi(a|h_t) \right] \quad (\pi \text{ depends only on } h_t)$$

$$= \sum_{t=1}^T \mathbb{E}_{s_t, h_t \sim d_t^\pi} \left[ \sum_{a_t} A^{\pi^E}(s_t, a_t) \pi(a|h_t) \right] \quad (\pi^E \text{ depends only on } s_t)$$

$$= \sum_{t=1}^T \mathbb{E}_{s_t, h_t \sim d_t^\pi} \left[ \sum_{a_t} A^{\pi^E}(s_t, a_t) \left( \pi(a|h_t) - \pi^E(a_t|s_t) \right) \right] \quad \text{as} \left( \sum_{a_t} A^{\pi^E}(s_t, a_t) \pi^E(a_t|s_t) = 0 \right) \tag{8}$$

Rearranging, we get the following form of PDL which we use in the main proof:

$$J(\pi^E) - J(\pi) = \sum_{t=1}^{T} \mathbb{E}_{s_t, h_t \sim d_t^\pi} \left[ \sum_{a_t} A^{\pi^E}(s_t, a_t) \left( \pi^E(a_t|s_t) - \pi(a_t|h_t) \right) \right] \tag{9}$$

The second result we use us from interactive imitation learning DAGGER (Ross et al., 2011) that reduces imitation learning to no-regret online learning. DAGGER shows that with $\pi^E$ as the expert teacher guarantees that after $N$ iterations, it will find at least one policy

$$\mathbb{E}_{s, h \sim d^\pi} ||\pi(.|h) - \pi^E(.|s)||_1 \leq \mathbb{E}_{s, h \sim d^\pi} ||\pi^\star(.|h) - \pi^E(.|s)||_1 + \gamma(N) \tag{10}$$

where $\gamma(N)$ is the average regret, and $d^\pi$ is the time average distribution of states and history induced by policy $\pi$.

We now prove the main result

$$\begin{aligned}
\text{AIG}(\pi, T) &= \frac{1}{T} \left( J(\pi^E) - J(\pi) \right) \\
&= \frac{1}{T} \sum_{t=1}^{T} \mathbb{E}_{s_t, h_t \sim d_t^\pi} \left[ \sum_{a_t} A^{\pi^E}(s_t, a_t) \left( \pi^E(a|s_t) - \pi(a|h_t) \right) \right] && \text{(From (9))} \\
&\leq ||A^{\pi^E}(.,.)||_\infty \frac{1}{T} \sum_{t=1}^{T} \mathbb{E}_{s_t, h_t \sim d_t^\pi} ||\pi(.|h_t) - \pi^E(.|s_t)||_1 && \text{(Holder's Inequality)} \\
&\leq H(\pi^E) \frac{1}{T} \sum_{t=1}^{T} \mathbb{E}_{s_t, h_t \sim d_t^\pi} ||\pi(.|h_t) - \pi^E(.|s_t)||_1 \\
&\leq H(\pi^E) \left( \frac{1}{T} \sum_{t=1}^{T} \mathbb{E}_{s_t, h_t \sim d_t^\pi} ||\pi^\star(.|h_t) - \pi^E(.|s_t)||_1 + \gamma(N) \right) && \text{From (10)} \\
&\leq H(\pi^E) \left( \epsilon(\pi^E, T) + \gamma(N) \right)
\end{aligned}$$

Rearranging, we get

$$\frac{1}{T} J(\pi) \geq \frac{1}{T} J(\pi^E) - H(\pi^E) \left( \epsilon(\pi^E, T) + \gamma(N) \right)$$

$\square$

To optimize the performance bound, we introduce a constrained privileged expert that provides actions similar to the privileged expert but is constrained to be close to realizable.

**Definition B.5** (Constrained Privileged Expert). Given a state $s_t$ and history $h_t$, a constrained privileged expert minimizes KL distance to the privileged expert while being in a $\delta$ KL ball of the non-privileged expert.

$$\pi_\delta^E(.|s_t, h_t) := \arg\min_{\tilde{\pi}} \text{KL} \left( \tilde{\pi}(.|s_t, h_t)||\pi^E(.|s_t) \right) \quad \text{s.t.} \quad \text{KL} \left( \tilde{\pi}(.|s_t, h_t)||\pi^E(.|h_t) \right) \leq \delta \tag{11}$$

We use the fact that we have a rich policy class $\Pi$ that includes the non-privileged expert $\pi^E(.|h)$, to show that the constrained privileged expert bounds the average realizability gap

$$\begin{aligned}
\epsilon(\pi_\delta^E, T) &= \sup_\pi \frac{1}{T} \sum_{t=1}^{T} \mathbb{E}_{s_t, h_t \sim d_t^\pi} ||\pi_\delta^E(a_t|s_t) - \pi^\star(a_t|h_t))||_1 \\
&\leq \sup_\pi \frac{1}{T} \sum_{t=1}^{T} \mathbb{E}_{s_t, h_t \sim d_t^\pi} \sqrt{2\text{KL}(\pi_\delta^E(a_t|s_t)||\pi^\star(a_t|h_t))} \\
&\leq \sup_\pi \frac{1}{T} \sum_{t=1}^{T} \mathbb{E}_{s_t, h_t \sim d_t^\pi} \sqrt{2\text{KL}(\pi_\delta^E(a_t|s_t)||\pi^E(a_t|h_t))} \quad \text{since } \pi^E(.|h_t) \in \Pi \\
&\leq \sqrt{2\delta} && \text{from (11)}
\end{aligned} \tag{12}$$

**Theorem B.6** (LEAP with Constrained Privileged Expert). *Running $N$ iterations of LEAP with constrained privileged expert $\pi_\delta^E$ yields at least one policy $\pi$ such that*

$$\frac{1}{T}J(\pi) \geq \frac{1}{T}J(\pi^E) - \left( \frac{1}{T}\underbrace{(J(\pi^E) - J(\pi_\delta^E))}_{\text{gap from privileged expert}} + H(\pi_\delta^E)\left( \underbrace{\sqrt{2\delta}}_{\text{realizability}} + \gamma(N) \right) \right) \quad (13)$$

*where $\pi^E$ is the privileged expert, $H(\pi_\delta^E)$ is the recoverability coefficient of $\pi_\delta^E$, $\delta$ is the KL constraint, $\gamma(N)$ is the average regret of the* DAGGER *update.*

The first term is the performance gap between the privileged and the constrained privileged expert. The second term is the realizability of the constrained privileged expert. As $\delta \to 0$, the realizability gap goes down while the performance gap and recoverability coefficient go up. There is a critical value of $\delta$ that achieves an optimal tradeoff.

*Proof.* The proof is similar to Theorem B.4 but with the modification that $\pi^E$ depends on both state and history.

The PDL in this case can be expressed as

$$J(\pi^E) - J(\pi) = \sum_{t=1}^{T} \mathbb{E}_{s_t, h_t \sim d_t^\pi} \left[ \sum_{a_t} A^{\pi^E}(s_t, h_t, a_t) \left( \pi^E(a_t|s_t, h_t) - \pi(a_t|h_t) \right) \right] \quad (14)$$

DAGGER (Ross et al., 2011) with $\pi_\delta^E$ as the expert teacher guarantees that after $N$ iterations, it will find at least one policy

$$\mathbb{E}_{s, h \sim d^\pi} ||\pi(.|h) - \pi_\delta^E(.|s, h)||_1 \leq \mathbb{E}_{s, h \sim d^\pi} ||\pi^\star(.|h) - \pi_\delta^E(.|s, h)||_1 + \gamma(N) \quad (15)$$

where $\gamma(N)$ is the average regret.

$$\begin{aligned}
\text{AIG}(\pi, T) &= \frac{1}{T}\left( J(\pi^E) - J(\pi) \right) \\
&= \frac{1}{T}\left( J(\pi^E) - J(\pi_\delta^E) \right) + \frac{1}{T}\left( J(\pi_\delta^E) - J(\pi) \right) \\
&= \frac{1}{T}\left( J(\pi^E) - J(\pi_\delta^E) \right) + \frac{1}{T}\sum_{t=1}^{T} \mathbb{E}_{s_t, h_t \sim d_t^\pi} \left[ \sum_{a_t} A^{\pi_\delta^E}(s_t, h_t, a_t) \left( \pi_\delta^E(a_t|s_t, h_t) - \pi(a_t|h_t) \right) \right] && \text{(From (14))} \\
&\leq \frac{1}{T}\left( J(\pi^E) - J(\pi_\delta^E) \right) + ||A^{\pi_\delta^E}(.,.,.)||_\infty \frac{1}{T}\sum_{t=1}^{T} \mathbb{E}_{s_t, h_t \sim d_t^\pi} ||\pi(.|h_t) - \pi_\delta^E(.|s_t, h_t)||_1 && \text{(Holder's Inequality)} \\
&\leq \frac{1}{T}\left( J(\pi^E) - J(\pi_\delta^E) \right) + H(\pi_\delta^E)\frac{1}{T}\sum_{t=1}^{T} \mathbb{E}_{s_t, h_t \sim d_t^\pi} ||\pi(.|h_t) - \pi_\delta^E(.|s_t, h_t)||_1 \\
&\leq \frac{1}{T}\left( J(\pi^E) - J(\pi_\delta^E) \right) + H(\pi_\delta^E)\left( \frac{1}{T}\sum_{t=1}^{T} \mathbb{E}_{s_t, h_t \sim d_t^\pi} ||\pi^\star(.|h_t) - \pi_\delta^E(.|s_t, h_t)||_1 + \gamma(N) \right) && \text{From (15)} \\
&\leq \frac{1}{T}\left( J(\pi^E) - J(\pi_\delta^E) \right) + H(\pi_\delta^E)\left( \epsilon(\pi_\delta^E, T) + \gamma(N) \right) \\
&\leq \frac{1}{T}\left( J(\pi^E) - J(\pi_\delta^E) \right) + H(\pi_\delta^E)\left( \sqrt{2\delta} + \gamma(N) \right) && \text{From (12)}
\end{aligned}$$

Rearranging, we get

$$\frac{1}{T}J(\pi) \geq \frac{1}{T}J(\pi^E) - \left( \frac{1}{T}\left( J(\pi^E) - J(\pi_\delta^E) \right) + H(\pi_\delta^E)\left( \sqrt{2\delta} + \gamma(N) \right) \right)$$

$\square$

### B.2 PRACTICAL APPROXIMATION FOR CONSTRAINED PRIVILEGED EXPERT

Writing out the Lagrangian of the constrained privileged expert (Def. B.5) and solving for the primal we get:

$$\pi^E(\rho, a|s, h) = \frac{1}{Z(s, h)} \pi^E(\rho, a_t|s_t) \left(\pi^E(\rho, a_t|h_t)\right)^\lambda \quad (16)$$

where $\lambda$ is the Lagrange multiplier.

For a fixed multiplier $\lambda$, this closed-form expression suggests a simple penalty-method approximation to constrained privileged expert:

1. Sample $N$ responses from both $\pi^E(\tilde{\rho}, \tilde{a}|\rho, a, h, s)$ and $\pi^E(\tilde{\rho}, \tilde{a}|\rho, a, h)$ to create $\mathcal{R} = \{\tilde{\rho}^i, \tilde{a}^i\}_{i=1}^N$

2. Reweigh the responses in $\mathcal{R}$ according to

$$\log \pi^E(\tilde{\rho}, \tilde{a}|\rho, a, h, s) + \lambda \log \pi^E(\tilde{\rho}, \tilde{a}|\rho, a, h) \quad (17)$$

3. Sample from the reweighed distribution.

Since the majority of our experiments involve closed-weights models (like `gpt-4o`), we are unable to access the logits of the expert. Instead, we design `LEAP` prompts to approximate the constrained privileged expert by providing instructions to not reveal the privileged information in the reasoning, and provide general reason and actions that should be predictable given only history $h_t$.

## C ADDITIONAL ABLATIONS

### C.1 LEAP VS SFT WITH PRIVILEGED INFORMATION

To investigate whether LEAP's performance gains primarily arise from the use of privileged information, we introduce a new baseline, SFT-privileged. This baseline collects demonstrations from a privileged teacher that has access to the same privileged information as LEAP and trains a student policy using Supervised Fine-Tuning (SFT). This follows the context distillation approach Snell et al. (2022), where privileged information is incorporated into the teacher's context and distilled into the student policy. To ensure a fair comparison, we use the same prompt structure as LEAP, preventing the teacher from explicitly revealing privileged information to the student.

**Key Observations:** (1) The SFT-privileged baseline achieves a success rate of 42.5%, which is significantly lower than LEAP (Iter 0: 65.7%, Iter 1: 91.0%). (2) When successful, SFT-privileged requires fewer actions (8.4) compared to LEAP (Iter 0: 12.7, Iter 1: 10.1).

| Model | %suc↑ | #act↓ | #act\|succ↓ |
|---|---|---|---|
| LEAP SFT-privileged | 42.5 | 20.8 | 8.4 |
| LEAP LEAP $\pi_0$ | 65.7 | 18.6 | 12.7 |
| LEAP LEAP $\pi_1$ | 91.0 | 11.9 | 10.1 |

Table 4: **Performance Comparison:** Success rate and action efficiency across models.

**Explanation:** At training time, the privileged teacher always provides successful demonstrations, offering general reasoning (e.g., "Cellphones are commonly found on desks") and using privileged information to locate objects and complete tasks. The SFT-privileged student is trained exclusively on these successful demonstrations. At test time, if the student's reasoning is correct, it successfully mimics the teacher with an optimal action count (8.4). Having been trained only on successful demonstrations, it lacks the ability to handle failure cases. For example, it might hallucinate incorrect objects or repeatedly visit the same location. This inability to recover leads to a significantly lower success rate (42.5%)

**Summary:** The improved performance of LEAP is driven not just by privileged information but by two key components: (1) On-policy corrections: On-policy corrections enable the student to recover from errors, as demonstrated by the significant improvement over the SFT-privileged baseline (91.0% vs. 42.5%). (2) Realizable corrections: Realizable corrections, where the teacher avoids revealing privileged information, also play a critical role in LEAP's performance. Ablations show a substantial drop when realizable corrections are removed (91.0% vs 5.2%).

| Model | Score↑ | #act↓ | r_att | r_option | r_type | r_price |
|-------|--------|-------|-------|----------|--------|---------|
| LEAP iter 0 | 29.4 | 21.1 | 28.0 | 25.8 | 35.0 | 36.8 |
| LEAP iter 1 | 36.8 | 18.7 | 36.6 | 33.0 | 44.2 | 45.2 |
| LEAP iter 2 | 39.8 | 18.4 | 39.9 | 31.7 | 48.1 | 51.0 |

Table 5: **Self-correction on WebShop:** LEAP self-improves a Llama3-8B model, where the teacher is a privileged version of the student.

| Model | %succ↑ | #act↓ | Pick | Clean | Heat | Cool | Look | Pick 2 |
|-------|--------|-------|------|-------|------|------|------|--------|
| LEAP ALFWORLD only iter0 | 65.7 | 18.6 | 66.7 | 74.2 | 73.9 | 66.7 | 66.7 | 35.3 |
| LEAP ALFWORLD only iter1 | 91.0 | 11.9 | 83.3 | 90.3 | 91.3 | 95.2 | 94.4 | 94.1 |
| LEAP COMBINED iter0 | 64.2 | 18.4 | 62.5 | 80.6 | 69.6 | 61.9 | 55.6 | 41.2 |
| LEAP COMBINED iter1 | 91.0 | 11.9 | 87.5 | 90.3 | 82.6 | 95.2 | 100.0 | 94.1 |

Table 6: **ALFWorld Results:** Success rates and per-task performance across different iterations.

| Model | Score↑ | #act↓ | r_att | r_option | r_type | r_price |
|-------|--------|-------|-------|----------|--------|---------|
| LEAP WEBSHOP only iter0 | 29.4 | 21.1 | 28.0 | 25.8 | 35.0 | 36.8 |
| LEAP WEBSHOP only iter1 | 51.8 | 16.7 | 60.0 | 16.1 | 73.4 | 78.4 |
| LEAP COMBINED iter0 | 21.3 | 22.6 | 16.6 | 29.7 | 38.4 | 21.5 |
| LEAP COMBINED iter1 | 37.9 | 20.1 | 42.9 | 17.1 | 52.2 | 56.8 |

Table 7: **WebShop Results:** Performance comparison across iterations for WebShop tasks.

## C.2 LEAP SELF-CORRECTION ON WEBSHOP

On WebShop, we observe that self-correction consistently improves performance across iterations, increasing from 29.4 to 36.8 and to 39.8 (Table 5). However, the performance gains are lower when using a Llama-8B teacher compared to a GPT-4o teacher. This is in line with our hypothesis that for environments like Webshop, where the privileged information is not as rich as ALFworld, the performance improvements do depend on the strength of the teacher to provide meaningful corrections.

## C.3 LEAP TRAINING ON ALL DATA

We run an experiment where we trained a single model on combined datasets (called COMBINED). To do so, we downsample ALFWorld dataset to be in the same order as Webshop. We run LEAP for two iterations and present results in Tables 6, 7 comparing to models trained on individual benchmarks.

We make the following observations: (a) LEAP shows similar magnitude of improvement when trained on all datasets compared to when trained on individual datasets (on ALFWorld 64.2 -> 91.0, on Webshop 21.3 -> 37.9). COMBINED model matches ALFWorld performance compared to a model trained on ALFWORLD only. (b) COMBINED model scores lower on Webshop compared to Webshop only. We attribute this to a mismatch in dataset size even after downsampling. However, the magnitude of improvement of LEAP is the same for both models.

## D EXPERIMENTAL DETAILS

### D.1 HYPER-PARAMETERS

#### D.1.1 TRAINING PARAMETERS

Tables 8 and 9 contain hyperparameters for SFT training and DPO/KTO training using LoRA for the different datasets. All training runs were on machines with either 2 or 4 RTX A6000 GPUs, each with 48 GB of memory per GPU.

| Dataset | AlfWorld | WebShop | InterCode |
|---|---|---|---|
| Model | Llama3-8B | Llama3-8B | Llama3-8B / Llama3-70B |
| Batch size | 64 | 16 | 16 |
| Max seq length | 8000 | 6000 | 6000 |
| Max epochs | 1 | 1 | 5 |
| Learning rate | 3e-5 | 3e-5 | 3e-5 |
| LoRA $\alpha$ | 64 | 64 | 64 |
| LoRA $r$ | 128 | 128 | 128 |
| LoRA dropout | 0.05 | 0.05 | 0.05 |
| Optimizer | AdamW | AdamW | AdamW |
| LR scheduler | cosine | cosine | cosine |

Table 8: Hyperparameters for SFT training using LoRA

| Method | DPO | KTO |
|---|---|---|
| Model | Llama3-8B | Llama3-8B |
| Batch size | 32 | 64 |
| Max seq length | 8000 | 8000 |
| Max prompt length | 6000 | 6000 |
| Max epochs | 1 | 1 |
| Regularization $\beta$ | 0.01 | 0.01 |
| Learning rate | 5e-7 | 5e-7 |
| LoRA $\alpha$ | 64 | 64 |
| LoRA $r$ | 128 | 128 |
| LoRA dropout | 0.05 | 0.05 |
| Optimizer | AdamW | AdamW |
| LR scheduler | cosine | cosine |

Table 9: Hyperparameters for preference training (DPO, KTO) using LoRA

### D.1.2 EVALUATION PARAMETERS

For inference using `Llama3-8b` and `Llama3-70b`, we use a temperature setting of 0.3 and a maximum token length of 256. The following code snippet shows the `.generate()` function used:

```
outputs = model.generate(
    tokenized_inputs["input_ids"],
    attention_mask=tokenized_inputs["attention_mask"],
    max_new_tokens=256,
    eos_token_id=[
        tokenizer.eos_token_id,
        tokenizer.convert_tokens_to_ids("<|eot_id|>"),
    ],
    temperature=0.3,
    pad_token_id=self.tokenizer.eos_token_id
)
```

Listing 1: Generation function with hyperparameters for Llama3 models

For all OpenAI model calls, including tasks such as generating expert teacher corrections, labeling reasoning for actions, or running prompt baselines like ReAct, we use the API call with the following hyperparameters. See Appendix E for the detailed prompts.

```
# model: gpt-4o
chat_completion = client.chat.completions.create(
    messages=messages,
    model="gpt-4o",
```

```
    temperature=0.3,
    top_p=1,
    n=1
)
response = chat_completion.choices[0].message.content

# model: gpt-4o-mini
chat_completion = client.chat.completions.create(
    messages=messages,
    model="gpt-4o-mini",
    temperature=0.3,
    top_p=1,
    n=1
)
response = chat_completion.choices[0].message.content
```

Listing 2: Hyperparameters for OpenAI GPT-4o and GPT-4o-mini models

## D.2 ALFWORLD

ALFWorld (Shridhar et al., 2020b) is a text-based game built on the ALFRED embodied benchmark (Shridhar et al., 2020a). Each task has a high-level objective, e.g. "heat mug and put it in cabinet" which the agent must complete by navigating and interacting with a virtual household through text commands, e.g. go to shelf 1, pick up mug 2, etc. It is a POMDP where the hidden state is the location of all objects, and the observations reveal only the objects the agent can see. To solve it, the agent must plan subgoals, track progress, and explore efficiently till time runs out. An interesting aspect of ALFWorld is the requirement to infer the likely locations of household objects (e.g. mugs are more likely to be on shelves or cabinets), which the agent can learn from interaction with the environment. There are 6 categories of tasks and two datasets: 139 in-distribution games and 134 out-of-distribution games. Time horizon is limited to 30.

We compare against ReAct (Yao et al., 2022b) with different base models and prompts. The original ReAct paper (Yao et al., 2022b) used text-davinci-002 with task-dependent in-context examples. We run ReAct wtug gpt-4o and gpt-4o-mini and an instruction prompt that is shared by all our methods. We also compare against BUTLER (Shridhar et al., 2020b), a seq2seq BERT model trained using DAGGER.[4]

We LoRA fine-tune Llama-3-8B model using gpt-4o as the expert teacher for 4 iterations. We use observation-action demonstrations on the training dataset of 3257 gamesShridhar et al. (2020b), which we annotate with gpt-4o to generate reasons and use this to train the BC policy $\pi_0$. We then roll out a policy and generate corrections from the privileged expert on unsuccessful rollouts. The correction data is then aggregated with the existing dataset and used to train a policy using SFT.

The privileged state in ALFworld consists of

1. Task: What is the overall task
2. Essential Objects: Enumerate all objects that are necessary to solve the game.
3. Critical Locations: List the important locations involved in the game solution.
4. Optimal Action Sequence: Suggest an optimal sequence of actions that could theoretically minimize the steps needed to complete the game, based on the log provided. Be sure to copy the action verbatim.

## D.3 WEBSHOP

WebShop (Yao et al., 2022a) is an online shopping environment consisting of 1.18 million real-world products and 12, 586 human-provided instructions. Each task is a shopping request from a user ("I need a gluten-free bacon, 4 oz pack of 2, priced under $50") that the agent has to solve through a

---

[4] Since we look at the single trial setting, it rules out approaches like Reflexion (Shinn et al., 2023) where agents require multiple trials on the same game.

set of interactions (search["gluten-free bacon"] or selecting options such as "4 oz pack of 2"). The metrics are the overall score, and 4 components of the score: whether the attributes match (ATT), whether the options match (OPT), whether the product types match (TYPE) and whether the price matches (PRICE). We evaluate on 500 test instructions, and limit time horizon to 30.

We compare against ReAct (Yao et al., 2022b) with different base models like GPT-4o and GPT-4o-mini. We use the same instruction prompt as in LEAP. We also compare against IL baseline in WebShop (Yao et al., 2022a) trained using BC. These baselines train two different selection models, one for selecting search queries from a list of options and one for clicking from a list of options. This constraint search space simplifies the problem.

We LoRA fine-tune Llama-3-8B model using GPT4-o as the expert teacher for 4 iterations. We initialize $\pi_0$ using observation-action demonstrations from Yao et al. (2022a) which we annotate with GPT4-o to generate reasons. We roll-out policies on the training dataset of 12086 tasks, and generate corrections from the privileged expert on unsuccessful rollouts. For every game in the training dataset, we construct privileged information that contains: the attributes the product must satisfy, the price constraint, the option that has to be selected, and an example product that would satisfy all criteria. The correction data is then aggregated with the existing dataset and used to train a policy using SFT.

The privileged state in ALFworld consists of

1. Task: What is the user instruction

2. Attributes: What are the attributes the product must have

3. Price: What price constraints must the product satisfy.

4. Example Product: Name of a product that satisfies all criteria.

## D.4 INTERCODE

Intercode Yang et al. (2024) is an interactive coding environment, where an agent interacts with a compiler/interpreter to execute its code and submit further refinements. We use the Bash environment, where a typical task is "Find all text files and write their names to a single file.", and the agent issues commands like `ls`, `find ..` to solve the task. Intercode builds on the NL2Bash (Lin et al., 2018) to create 4 interactive datasets. We use the first 2 for training, and the next 2 for testing. We limit the time horizon to 10. We compare against ReAct Yao et al. (2022b) with `gpt-4o` and `gpt-4o-mini`. We also compare against the top 3 entries on the Intercode leaderboard, which are the `TryAgain` baselines from Yang et al. (2024) with `gpt-4`, `gpt-3.5` and CodeLlama-34B-INST respectively. Note the numbers are directly from the leaderboard which evaluates on all (train+test) data and, while our numbers are only on test.

## E PROMPTS

We define the following three prompts across all datasets:

1. **Expert corrections template:** This prompt is used by the LLM expert teacher to provide corrections on the student agent's trajectories. These corrections offer improved reasoning and actions that balance the trade-off between utilizing the privileged state (accessible only to the expert during training) and ensuring the corrections are realizable by the student agent.
2. **Student agent template:** This prompt is used during test time by the Llama3 student agent models to generate reason and action trajectories that achieve the task objective. It is also used for prompting baselines, such as GPT-4 or ReAct, which generate reasoning and action trajectories to achieve the task objectives.
3. **Autolabel reasons template:** For datasets that contain human demonstrations with action labels (like AlfWorld, WebShop), this prompt is used to generate reasoning annotations for actions that the human took. Generating reasons is necessary as the student agent needs to be trained to produce both reason and actions in its output trajectories.

For prompts used in different ablation studies, please refer to the accompanying code.

## E.1 ALFWORLD

### E.1.1 EXPERT CORRECTIONS TEMPLATE

```
{% if system %}
You are given a trajectory containing observations, reason and actions
    generated by a student agent solving a text world game. You are a
    teacher who has access to a "privileged state" that contains secret
     information sufficient to solve the game but is hidden from the
    student. Your goal is to improve how the student solves the game by
     improving their reason and action at every timestep.

### Input
You will be provided with a JSON file that logs the student agent's
    observation, candidate_actions, reason and action at every timestep
     while playing a text-based game. The student fails to solve the
    task within the time horizon of the game.

The structure is an array of objects, containing the following at each
    timestep:
- timestep: Index of the current timestep
- observation: The observation provided to the student agent
- candidate_actions: The set of allowed actions
- reason: The reason generated by the student agent to justify their
    action
- action: The action taken by the student agent

You will also be provided the privileged state that contains hidden
    information that specifies how to solve the task.

### Task
* Analyze the student trajectory and summarize the mistakes it is
    making when trying to solve the game
* Refer to your privileged state to know how the game can be solved
* Generate IMPROVED reason and action for the student at every timestep
     to guide them towards the goal
* Base your improved reasons solely on the student's historical
    observations and actions up to each timestep
* Do NOT include any information from your privileged state in the
    improved reasons, as the student does not have access to those
* Offer GENERAL principles or hints in your improved reasons that
    explain why the student should prefer your suggested action over
    their original action. This would help the student generalize
    better.
* When generating improved reason, action at timestep t, assume that
    the student has followed their original trajectory up until
    timestep t. Copy over the original observation at timestep t from
    the student trajectory.
* When generating improved action at timestep t, make sure the action
    is available in the candidate_actions at timestep t. Don't select
    an action that is not available.

Important: Provide GENERAL principles or hints in your improved reasons
     that explain why the student should prefer your suggested action
    over their original action.
* If you know from your privileged state that an object is in a
    different location from where the student is exploring, use common
    sense rationale to guide the student
* Do not directly instruct the student to go to the desired location,
    as they do not have access to the privileged information you
    possess
* For example, if the student is exploring the wrong area, instead of
    stating the object is in a different location, suggest a general
```

```
    principle like, "It might be useful to explore areas because ... ",
     ".. this item is often found in such places", etc
* By following these steps, you help the student understand the logic
    behind the actions without revealing privileged information
* Also note that you cannot carry more than one item at a time

### Output
The output is a JSON containing a summary and a trajectory with the
    same length as the input student trajectory as follows:
```json
{
    "summary": your summary of the mistakes the student is making,
    "trajectory": [
    {
        "timestep": Index of the current timestep,
        "original_observation": The original observation made by
    student agent at timestep t (copy as is from student trajectory),
        "original_reason": The original reason generated by the student
     at timestep t (copy as is from student trajectory),
        "original_action": The original action the student took at
    timestep t (copy as is from student trajectory),
        "corrected_reason": The corrected reason that the student
    should generate,
        "corrected_action": The corrected action that the student
    should take (chosen from list of candidate_actions at timestep t)
    },
    {
        ...
    }
    ...
    ]
}
```
{% endif %}
{% if not system %}
The student trajectory is below:
{{student_trajectory}}

The privileged state for the task is below:
{{privileged_state}}

Provide the ### Output in the JSON format specified above.
{% endif %}
"""
```

### E.1.2 STUDENT AGENT TEMPLATE

```
{% if mode == 'input' %}
You are an intelligent assistant named ALFRED in a text-based
    interactive game called TextWorld. Your objective is to complete
    the given tasks by reasoning through the information provided and
    taking appropriate actions.

Your task is the following:
{{task}}

Below is the history of previous observations and actions:
{{ observation_action_history }}

Given the history of previous observation and action above, a reminder
    that your task is:
```

```
{{task}}

You are given as input the current observation and the list of possible
    candidate_actions:
{
    "observation": {{observation}},
    "candidate_actions": {{candidate_actions}}
}

Your goal is to generate the action to take at this time step (chosen
    from candidate_actions) along with the reason for taking the action
    .

Please follow these general instructions:
* You MUST choose action from the list of candidate_actions.
* If "observation": "Nothing happens.", it is because you chose an
    invalid action not from the list of candidate_actions in the
    previous timestep.
* Oftentimes the task requires you interact with objects not present in
    your observation. You must search the environment to locate the
    objective.
* Consult the history of previous observations and actions to see what
    actions you have tried already so as to not repeat your actions.
* Do NOT repeat the same action as the last action in your
    observation_action_history. It's going to yield the same result.
* Make sure action is VERBATIM copied from the list of
    candidate_actions.

You need to generate a response in the following format. Please issue
    only a single action at a time.
REASON:
Rationale for what action to take next based on the task and previous
    history. In your reason, consult candidate_actions to precisely
    state VERBATIM which action you will do.
ACTION:
The action to be taken, chosen ONLY from candidate_actions
{% elif mode == 'output' %}
REASON:
{{ reason }}
ACTION:
{{ action }}
{% endif %}
```

### E.1.3 AUTOLABEL REASONS TEMPLATE

```
{% if system %}
You are given a JSON file containing observations and actions generated
    by an expert agent solving text world games. Your goal is to
    generate the reasoning process of the agent to justify each of
    their actions.

### Input
The input is a JSON array of objects, each containing:
- 'observation': The state observed by the agent at a specific timestep
    .
- 'action': The action taken by the agent in response to the
    observation.

Example input format:
'''json
[
```

```
    {
        "observation": "You see a locked door.",
        "action": "Use key on door."
    },
    ...
]
```

### Output
The output should be a JSON array of objects, each containing:
- `observation`: The state observed by the agent at a specific timestep
  .
- `reason`: The reasoning behind the action taken by the agent.
- `action`: The action taken by the agent in response to the
    observation.

Example output format:
```json
[
    {
        "observation": "You see a locked door.",
        "reason": "The door is locked and I have a key that might open
    it.",
        "action": "Use key on door."
    },
    ...
]
```

### Requirements
1. **Causal Constraints**: Ensure the reasoning satisfies causal
    constraints. Use only the observations up to the current timestep
    to justify the action. Do not use future information.
2. **Brevity and Relevance**: The reasoning should be brief and
    relevant, using the observations up until that timestep to justify
    the action.
3. **Sequence Consistency**: Ensure the output maintains the same
    sequence of observations and actions as in the input.

{% endif %}

{% if not system %}
The input JSON file is below:
{{input}}

Provide the output in the format specified above.
{% endif %}
```

## E.2   WEBSHOP

### E.2.1   EXPERT CORRECTIONS TEMPLATE

```
{% if system %}
You are given a trajectory containing observations, reason and actions
    generated by a student agent solving a shopping task by searching
    for items in a website and clicking on links. You are a teacher who
     has access to a "privileged state" that contains secret
    information sufficient to solve the task but is hidden from the
    student. Your goal is to improve how the student solves the task by
     improving their reason and action at every timestep.

### Input
```

```
You will be provided with a JSON file that logs the student agent's
    observation, candidate_actions, reason and action at every timestep
    while solving a shopping task. The student fails to solve the task
    within the time horizon of the task.

The structure is an array of objects, containing the following at each
    timestep:
- timestep: Index of the current timestep
- observation: The observation provided to the student agent
- candidate_actions: The set of allowed actions
- reason: The reason generated by the student agent to justify their
    action
- action: The action taken by the student agent

You will also be provided the privileged state that contains hidden
    information that specifies how to solve the task.

### Task
* Analyze the student trajectory and summarize the main mistakes it is
    making when trying to solve the task
* Refer to your privileged state to know how the task can be solved
* Correct a select set of timesteps where you know the student was
    definitely doing a wrong action
* Generate IMPROVED reason and action for the student at every timestep
    to guide them towards the goal
* Base your improved reasons solely on the student's historical
    observations and actions up to each timestep
* Do NOT include any information from your privileged state in the
    improved reasons, as the student does not have access to those
* Offer GENERAL principles or hints in your improved reasons that
    explain why the student should prefer your suggested action over
    their original action. This would help the student generalize
    better.
* When generating improved reason, action at timestep t, assume that
    the student has followed their original trajectory up until
    timestep t. Copy over the original observation at timestep t from
    the student trajectory.
* When generating improved action at timestep t, make sure the action
    is available in the candidate_actions at timestep t. Don't select
    an action that is not available.  If candidate_actions has only
    click actions, you can only choose a click action from the list. If
    candidate_actions has only search[<search query>], you must search
    by generating the search keywords on your own.

Important:
(1) Provide GENERAL principles or hints in your improved reasons that
    explain why the student should prefer your suggested action over
    their original action.
* If you know from your privileged state that the desired product is
    different from the one the student is considering, use common sense
    rationale to guide the student
* Do not directly instruct the student to search the ground truth
    product, as they do not have access to the privileged information
    you possess
* By following these steps, you help the student understand the logic
    behind the actions without revealing privileged information

(2) Put corrections sparingly.
* Be strategic about which timesteps you want to correct.
* If you think an action is good enough or reasonable, don't bother
    correcting. Just copy over the original reason and action.

(3) Correct indecisive behavior of the student agent
* Often times, you will see the student agent not finishing the task
    and instead continually browsing through items.
```

```
* Provide corrections at key moments to help it resolve indecisiveness.
* It's better to purchase a suboptimal item than not finish the task
    within the time horizon.
* The student gets partial points for matching a subset of the criteria
     in the instruction. They get 0 points for not finishing the task
    in the time horizon (max timesteps in student trajectory)

(4) Considerations for search.
* Specifying prices in search does not work. Other specifications are
    fine.
* Only correct egregious failures in search queries, e.g. searching for
     a specification not mentioned in the instruction. Otherwise search
     usually does not require any correction.

(5) Considerations for click.
* When suggesting corrected_action, make sure this action exists in the
     list of candidate_actions at that timestep.

### Output
The output is a JSON containing a summary and a trajectory with the
    same length as the input student trajectory as follows:
```json
{
    "summary": your summary of the mistakes the student is making,
    "trajectory": [
    {
        "timestep": Index of the current timestep,
        "is_corrected": True/False depending on whether this timestep
    is corrected or not
        "corrected_reason": The corrected reason that the student
    should generate. If is_corrected=False, copy over original reason.
    ,
        "corrected_action": The corrected action that the student
    should take (chosen from list of candidate_actions at timestep t).
    If is_corrected=False, copy over original action. If click action,
    make sure corrected_action belongs to list of candidate_actions at
    the corresponding timestep in input student_trajectory.
    },
    {
        ...
    }
    ...
    ]
}
```
{% endif %}
{% if not system %}
The student trajectory is below:
{{student_trajectory}}

The privileged state for the task is below:
{{privileged_state}}

Provide the ### Output in the JSON format specified above.
{% endif %}
```

### E.2.2 STUDENT AGENT TEMPLATE

```
{% if mode == 'input' %}
You are an intelligent shopping agent. Your objective is to solve
    shopping tasks by searching for items in a website, clicking on
    links until you solve the task.
```

```
Below is the history of previous observations and actions:
{{ observation_action_history }}

You are given as input the current observation that contains
    instructions for the task and contents of the current webpage. You
    are also given a list of possible candidate_actions:
{
    "observation": {{observation}},
    "candidate_actions": {{candidate_actions}}
}

Your goal is to generate the action to take at this time step along
    with the reason for taking the action.

Please follow these general instructions:
* You MUST choose action from the list of candidate_actions. If
    candidate_actions has only click actions, you can only choose a
    click action from the list. If candidate_actions has only search[<
    search query>], you must search by generating the search keywords
    on your own.
* When choosing a search action, you MUST follow the format search[<
    search query>]
* Choose search keywords that are informative but not too specific or
    too broad. Examples of search:
search[6 foot coaxial cable, pack of 3]
search[satin brass frosted hallway light fixture]
* When choosing click action, make sure action is VERBATIM copied from
    the list of candidate_actions. Examples of click:
click[satin brass | frosted]
click[buy now]
* Often times the task requires you to find objects not in the current
    webpage. You must search the webpages to locate the object and
    verify it indeed matches the specifications in the instructions. If
    not, go back and refine your search.
* Consult the history of previous observations and actions to see what
    actions you have tried already so as to not repeat your actions.
* Do NOT repeat the same action as the last action in your
    observation_action_history. It's going to yield the same result.
* When choosing click action, make sure action is VERBATIM copied from
    the list of candidate_actions.
* Important: You must try to finish the task and click on buy now as
    quickly as possible. It's better to purchase a suboptimal item than
     not finish the task within the time horizon. You get partial
    points for matching a subset of the criteria in the instruction.
    You get 0 points for not finishing the task in the time horizon (
    typically 20 timesteps).

You need to generate a response in the following format. Please issue
    only a single action at a time.
REASON:
Rationale for what action to take next based on the task and previous
    history. In your reason, consult candidate_actions to precisely
    state VERBATIM which action you will do.
ACTION:
The action to be taken, chosen ONLY from candidate_actions
{% elif mode == 'output' %}
REASON:
{{ reason }}
ACTION:
{{ action }}
{% endif %}
```

### E.2.3 AUTOLABEL REASONS TEMPLATE

```
{% if system %}
You are given a trajectory containing observations and actions
    generated by an agent solving a web shopping task. Your goal is to
    generate the reasoning process that the agent must have been
    thinking at each timestep to justify each of their actions.

### Input
You will be provided with a JSON file that logs the agent's observation
     and action at every timestep while solving the web shop task.

The structure is an array of objects, containing the following at each
    timestep:
- timestep: Index of the current timestep
- observation: The instruction the agent has to solve and the current
    webpage observed by the agent at a specific timestep.
- action: The action taken by the agent.

### Task
* Analyze the agent trajectory
* Generate reason for the agent at every timestep. The reason should be
     in first person, e.g. "I should .."
* Base your generated reason solely on the historical observations and
    actions up to each timestep
* Offer GENERAL principles or hints in your reasons that explain why
    the agent took the action
* For example, if the agent is backtracking by clicking on previous,
    explain why that is the case
* Do NOT include any information from future observations as the agent
    does not know the future at that time
* Ensure the output maintains the same sequence of actions as in the
    input.
* In your REASON, be sure to provide detailed justification for the
    action. If candidate_actions has only click actions, specify which
    option from the list are you choosing as your action.
* Often times, the task requires you to select both a product and
    specific options for that product. Be sure to justify why you are
    selecting a particular option for a product by referring to the
    task in your observation.
* Often times, the agent settles for an option even though it may not
    be exactly optimal. This is because the agent has a fixed time
    horizon within which they must click buy now. Ensure that your
    reason rationalizes why it's good to finish the task by clicking on
     buy now.

### Output
The output is a trajectory represented as a JSON array of objects with
    the same length as the input trajectory as follows:
```json
[
    {
        "timestep": Index of the current timestep,
        "reason": A reason to justify the action given observation and
    action up until the current timestep
        "action": The action taken by the agent (copy as is from input
    trajectory)
    },
    ...
]
{% endif %}
{% if not system %}
The input trajectory is below:
```

```
{{input}}

Provide the ### Output in the JSON format specified above.
{% endif %}
```

### E.3  INTERCODE

### E.3.1  EXPERT CORRECTIONS TEMPLATE

```
{% if system %}
You are given a trajectory containing observations, reason and actions
    generated by a student agent interacting with a MySQL Database
    system using sql queries to answer a question.  You are a teacher
    who has access to a "privileged state" that contains secret
    information sufficient to solve the task but is hidden from the
    student. Your goal is to improve how the student solves the task by
     improving their reason and action at every timestep.

### Input
You will be provided with a JSON file that logs the student agent's
    observation, reason and action at every timestep while solving a
    shopping task. The student fails to solve the task within the time
    horizon of the task.

The structure is an array of objects, containing the following at each
    timestep:
- timestep: Index of the current timestep
- observation: The observation provided to the student agent
- reason: The reason generated by the student agent to justify their
    action
- action: The action taken by the student agent

You will also be provided the privileged state. The privileged state
    contains an example of an optimal SQL command to solve the task and
     the expected observation that solves the task. Note that it is the
     observation that the student gets evaluated on. Multiple possible
    actions can lead to that observation.

### Task
* Analyze the student trajectory and summarize the main mistakes the
    student is making when trying to solve the task.
* Refer to your privileged state to understand how the task can be
    solved optimally.
* Correct a select set of timesteps where the student's action is
    clearly wrong.
* Generate **IMPROVED REASONS** and **ACTIONS** for the student at each
     timestep to guide them towards the goal. The improved reason
    should be framed as if it is the student's new reasoning, e.g., "I
    should ...".
* The improved reason and action are intended to replace the student's
    original reason and action at that timestep. Do NOT refer to the
    original incorrect reason and action when generating improved
    reason and action.
* Base your improved reasons solely on the student's historical
    observations and actions up to that timestep.
* Do NOT include any information from your privileged state in the
    improved reasons, as the student does not have access to that
    information.
* Offer general principles or hints in your improved reasons that
    explain why the student should prefer your suggested action over
    their original action. This would help the student generalize
    better.
```

```
* When generating an improved reason and action at timestep `t`, assume
    the student followed their original trajectory up until timestep `
    t`.

Important:
(1) Provide **GENERAL principles or hints** in your improved reasons.
* If you know from your privileged state that a particular set of
    tables should be queried to solve the task, suggest reasons and
    actions that guide the student to first discover those tables.
* Do not directly instruct the student to output the privileged action,
     as they do not have access to privileged information.
* By following these steps, you help the student understand the logic
    behind the actions without revealing privileged information.

(2) Apply corrections sparingly.
* Be strategic about which timesteps you choose to correct.

### Output
The output is a JSON containing a summary and a trajectory with the
    same length as the input student trajectory as follows:
```json
{
    "summary": your summary of the mistakes the student is making,
    "trajectory": [
    {
        "timestep": Index of the current timestep,
        "original_reason": The original reason generated by the student
     at timestep t (copy as is from student trajectory),
        "original_action": The original action the student took at
    timestep t (copy as is from student trajectory),
        "is_corrected": True/False depending on whether this timestep
    is corrected or not,
        "corrected_reason": The corrected reason that the student
    should generate at timestep t. If is_corrected=False, leave blank ,
        "corrected_action": The corrected action that the student
    should take at timestep t. If is_corrected=False, leave blank
    },
    {
        ...
    }
    ...
    ]
}
```
{% endif %}
{% if not system %}
The student trajectory is below:
{{student_trajectory}}

The privileged state for the task is below:
{{privileged_state}}

Provide the ### Output in the JSON format specified above.
{% endif %}
```

### E.3.2  STUDENT AGENT TEMPLATE

```
{% if system %}
You are given a trajectory containing observations, reason and actions
    generated by a student agent interacting with a MySQL Database
    system using sql queries to answer a question.  You are a teacher
```

```
        who has access to a "privileged state" that contains secret
        information sufficient to solve the task but is hidden from the
        student. Your goal is to improve how the student solves the task by
         improving their reason and action at every timestep.

### Input
You will be provided with a JSON file that logs the student agent's
    observation, reason and action at every timestep while solving a
    shopping task. The student fails to solve the task within the time
    horizon of the task.

The structure is an array of objects, containing the following at each
    timestep:
- timestep: Index of the current timestep
- observation: The observation provided to the student agent
- reason: The reason generated by the student agent to justify their
    action
- action: The action taken by the student agent

You will also be provided the privileged state. The privileged state
    contains an example of an optimal SQL command to solve the task and
     the expected observation that solves the task. Note that it is the
     observation that the student gets evaluated on. Multiple possible
    actions can lead to that observation.

### Task
* Analyze the student trajectory and summarize the main mistakes the
    student is making when trying to solve the task.
* Refer to your privileged state to understand how the task can be
    solved optimally.
* Correct a select set of timesteps where the student's action is
    clearly wrong.
* Generate **IMPROVED REASONS** and **ACTIONS** for the student at each
     timestep to guide them towards the goal. The improved reason
    should be framed as if it is the student's new reasoning, e.g., "I
    should ...".
* The improved reason and action are intended to replace the student's
    original reason and action at that timestep. Do NOT refer to the
    original incorrect reason and action when generating improved
    reason and action.
* Base your improved reasons solely on the student's historical
    observations and actions up to that timestep.
* Do NOT include any information from your privileged state in the
    improved reasons, as the student does not have access to that
    information.
* Offer general principles or hints in your improved reasons that
    explain why the student should prefer your suggested action over
    their original action. This would help the student generalize
    better.
* When generating an improved reason and action at timestep 't', assume
     the student followed their original trajectory up until timestep '
    t'.

Important:
(1) Provide **GENERAL principles or hints** in your improved reasons.
* If you know from your privileged state that a particular set of
    tables should be queried to solve the task, suggest reasons and
    actions that guide the student to first discover those tables.
* Do not directly instruct the student to output the privileged action,
     as they do not have access to privileged information.
* By following these steps, you help the student understand the logic
    behind the actions without revealing privileged information.

(2) Apply corrections sparingly.
* Be strategic about which timesteps you choose to correct.
```

```
### Output
The output is a JSON containing a summary and a trajectory with the
    same length as the input student trajectory as follows:
```json
{
    "summary": your summary of the mistakes the student is making,
    "trajectory": [
    {
        "timestep": Index of the current timestep,
        "original_reason": The original reason generated by the student
     at timestep t (copy as is from student trajectory),
        "original_action": The original action the student took at
    timestep t (copy as is from student trajectory),
        "is_corrected": True/False depending on whether this timestep
    is corrected or not,
        "corrected_reason": The corrected reason that the student
    should generate at timestep t. If is_corrected=False, leave blank ,
        "corrected_action": The corrected action that the student
    should take at timestep t. If is_corrected=False, leave blank
    },
    {
        ...
    }
    ...
    ]
}
```
{% endif %}
{% if not system %}
The student trajectory is below:
{{student_trajectory}}

The privileged state for the task is below:
{{privileged_state}}

Provide the ### Output in the JSON format specified above.
{% endif %}
```

