# OpenReview forum: "Better than Your Teacher: LLM Agents that learn from Privileged AI Feedback"
_ICLR.cc/2025/Conference — ICLR 2025 Poster_

### Official Review · Reviewer_Yn4v · 2024-10-21

**Soundness:** 3
**Presentation:** 3
**Contribution:** 3
**Rating:** 6
**Confidence:** 3

**Summary:**

The authors propose LEAP, an iterative fine-tuning framework that continually improves LLM agents using feedback from AI expert teachers. Notably, they equip the expert teachers with a privileged state. They evaluate LEAP on several decision-making benchmarks, and their experiments show that LEAP enables weak student models to exceed the performance of strong teacher models.

**Strengths:**

1. Novel methods: There have been a huge amount of research papers discussing about LLM self-improvement. However, the idea of equipping the expert teachers with a privileged state is relatively novel.

2. Good experimental results: LEAP significantly outperforms all baselines, even the stronger teachers like GPT-4o.

3. Diverse experiments: The authors conduct various experiments across several fundamentally different benchmarks.

**Weaknesses:**

1. Privileged states can be hard to get in real world environments (though this may not fit in the scope of this work).

2. Training an agent for each benchmark seperately is specific. The results would be much more promising if the authors demonstrate the LLM can be finetuned on all benchmarks together and still achieve good results.

3. Some minor typos: For example, in line 24, GPT4-o should be written as GPT-4o. In line 127, $\rho_t^*,a_t^*$ should be $(\rho_t^*,a_t^*)$.

**Questions:**

1. Have the author tried to perform preference optimization after SFT? Will this lead to a even higher score?

2. The authors admit that generating interaction rollouts is time-consuming. How long does it typically take in terms of magnitude—hours, days, or even longer?

---

> ### Author Response · Authors · 2024-11-17
>
> Thank you for your thoughtful evaluation and constructive feedback!
>
> **"Q. Privileged states can be hard to get in real world environments (though this may not fit in the scope of this work)."**
>
> While we agree that privileged information may appear unavailable in real-world scenarios, it is often accessible and can be easily derived from existing sources. In our three evaluation domains, we extract privileged information from pre-existing data, requiring no additional data collection effort.
>
> In particular, one can think of three common sources of privileged information for any domain:
> * **Simulator States.** Available in most agentic domains requiring interaction with simulated environments before real-world deployment. For instance, in AlfWorld, while agents observe only their current location, the simulator provides full item locations as privileged information during training.
> * **Evaluation Criteria.** Metrics or success criteria are available in nearly every domain. For example, in WebShop we use the product attributes, options, target prices, and in InterCode we use the goal commands and unit tests.
> * **Human Annotations.** While not universally available, they can often be derived from existing human demonstrations. For example, in AlfWorld, we extract key subgoal information (e.g., "drawer 3 contains book 2") from human demonstrations. Similarly, dialog agents often rely on conversation-level subgoal annotations. In fact, subgoal annotations are significantly easier to collect compared to dense demonstrations, making them a practical alternative as long as approaches (like LEAP) can use such information.
>
> **"Q. The results would be much more promising if the authors demonstrate the LLM can be finetuned on all benchmarks together and still achieve good results."**
>
> Thanks for the suggestion. Our primary goal was to demonstrate generalization across tasks within each benchmark, as each benchmark contains a large number of diverse tasks (e.g. >10k for WebShop, >3k for AlfWorld). The benchmarks themselves are quite distinct, for example, text-based games, web navigation, and interactive coding, with very little overlap between them. As a result, training across benchmarks may not provide significant benefits and could potentially exceed the capacity of an 8B model.
>
> That said, we are happy to run this experiment and will share the results here once it is completed.
>
> **"Q. Some minor typos: For example, in line 24, GPT4-o should be written as GPT-4o .."**
>
> Thanks for pointing this out. We have fixed these typos and updated the draft on OpenReview.
>
> **Questions**
>
> **"Q1. Have the author tried to perform preference optimization after SFT? Will this lead to a even higher score?"**
>
> In the paper, we report results for the following approaches:
>
> *1. SFT with dataset aggregation.* This approach involves fine-tuning on aggregated datasets iteratively.
>
> SFT on (iter 0 demonstrations) → SFT on (iter 0 demonstrations + iter 1 corrections) → SFT on (iter 0 demonstrations + iter 1 corrections + iter 2 corrections)
>
> *2. SFT + DPO using corrections as preferences.* Here, corrections are formatted as preference pairs (original, improved) and used for DPO and DPO is regularized to the model from the previous iteration.
>
> SFT on (iter 0 demonstrations) → DPO on (iter 1 corrections) → DPO on (iter 2 corrections)
>
> We also tested a more complex sequence in a smaller-scale experiment:
>
> SFT on (iter 0 demonstrations) → SFT on (iter 0 demonstrations + iter 1 preferences) → DPO on (iter 1 preferences) → SFT on (iter 0 demonstrations + iter 1 corrections + iter 2 corrections) → DPO on (iter 2 preferences)
>
> However, this additional DPO step did not lead to noticeable improvements. The SFT step on corrections alone was effective, and further optimization with DPO did not significantly improve the results.
>
> **"Q2. The authors admit that generating interaction rollouts is time-consuming. How long does it typically take in terms of magnitude—hours, days, or even longer?"**
>
> Generating interaction rollouts is time consuming primarily due to the large number of training environments, as corrections are being collected on the training set tasks. For a dataset like AlfWorld, each rollout takes approximately 30s, and with 3000 training examples, the total time is roughly ~25 hr (=30 * 3000).
>
> Rollouts can be significantly sped up through parallelization, though they are not trivially batchable since each rollout requires its own environment instance and a separate history of observations and actions. To address this, we provide code to host the model we are rolling out on a local SGLang server. This setup allows multiple environment instances to run in parallel, with each instance independently querying the model server, enabling efficient parallel rollout generation and significantly reducing the overall time.

---

> > ### Author Response · Authors · 2024-11-22
> >
> > **"Q. The results would be much more promising if the authors demonstrate the LLM can be finetuned on all benchmarks together and still achieve good results."**
> >
> > Following up on this, we ran an experiment where we trained a single model on combined datasets (called COMBINED). To do so, we downsample ALFWorld dataset to be in the same order as Webshop. We run LEAP for two iterations and present results below comparing to models trained on individual benchmarks
> >
> > **ALFWorld Results**
> > | Model                  | Success Rate | Num actions | Pick  | Clean | Heat  | Cool  | Look  | Pick 2 |
> > |------------------------|--------------|-------------|-------|-------|-------|-------|-------|--------|
> > | ALFWORLD only iter0    | 65.7         | 18.6        | 66.7  | 74.2  | 73.9  | 66.7  | 66.7  | 35.3   |
> > | ALFWORLD only iter1    | 91.0         | 11.9        | 83.3  | 90.3  | 91.3  | 95.2  | 94.4  | 94.1   |
> > |                      |       |             |       |          |        |         |
> > | COMBINED iter0         | 64.2         | 18.4        | 62.5  | 80.6  | 69.6  | 61.9  | 55.6  | 41.2   |
> > | COMBINED iter1         | 91.0         | 11.9        | 87.5  | 90.3  | 82.6  | 95.2  | 100   | 94.1   |
> >
> > **Webshop Results**
> > | Model                | Score | Num actions | r_att | r_option | r_type | r_price |
> > |----------------------|-------|-------------|-------|----------|--------|---------|
> > | WEBSHOP only iter0   | 29.4  | 21.1        | 28    | 25.8     | 35     | 36.8    |
> > | WEBSHOP only iter 1  | 51.8  | 16.7        | 60    | 16.1     | 73.4   | 78.4    |
> > |                      |       |             |       |          |        |         |
> > | COMBINED iter0       | 21.3  | 22.6        | 16.6  | 29.7     | 38.4   | 21.5    |
> > | COMBINED iter 1      | 37.9  | 20.1        | 42.9  | 17.1     | 52.2   | 56.8    |
> >
> > We make the following observations:
> > * LEAP shows similar magnitude of improvement when trained on all datasets vs on individual datasets (on ALFWorld 64.2 -> 91.0, on Webshop 21.3 -> 37.9).
> > * COMBINED model matches ALFWorld performance compared to a model trained on ALFWORLD only.
> > * COMBINED model scores lower on Webshop compared to Webshop only. We attribute this to a mismatch in dataset size even after downsampling. However, the magnitude of improvement of LEAP is the same for both models.
> >
> > We plan to complete these experiments, repeat them for a couple of different data mix options, and add the results to the appendix of the final draft.

---

> > > ### Comment · Reviewer_Yn4v · 2024-11-23
> > >
> > > Thanks for your detailed responses! They have mostly addressed my concerns. In particular, the supplemental experiments show that while the initial performance in the combined training setup is understandably lower, the similar magnitude of improvement demonstrates the generalizability of the proposed approach.
> > >
> > > Additionally, I appreciate the authors’ reference to the sim-to-real setting in robotics, as discussed in response to Reviewer nN6Y. Using privileged states during training in a simulator, followed by deployment in real-world environments without them, is indeed a well-established practice. While sim-to-real often involves more complex training techniques and (perhaps) the handling of visual inputs, this work seems to offer a good insight.

---

> ### Author Response · Authors · 2024-11-25
>
> Thank you so much for taking the time to review our work and helping improve it with your feedback. We really appreciate it!

---

### Official Review · Reviewer_nN6Y · 2024-10-27

**Soundness:** 2
**Presentation:** 2
**Contribution:** 3
**Rating:** 6
**Confidence:** 3

**Summary:**

The paper seeks to address LLM Agents' difficulties when it comes from recovering from errors during task execution.

They begin by framing decision-making as a Partially Observable Markov Decision Process (POMDP).
Given this framing, they approach the issue by proposing LEAP, a DAGGER-like iterative training framework in which an expert teacher is equipped with "privileged state" corrects the faulty student agent trajectories during training.

Here, the "privileged state" is simply a representation of the environment state that contains more information than what is available to the student.

After outlining the problem and LEAP, the authors briefly examine their setup, analytically showing that there is a tradeoff between the performance and realisability of the privileged expert teacher.

The paper then moves on to the experiments, where the authors test their framework against a number of baselines across a variety of domains (text, web agents, and coding), demonstrating improvements in performance. Their experiments also show the possibility of student agents surpassing their teachers and of self-improvement when using a privileged version of themselves as the teacher. The authors also experimentally explore the tradeoff between how much privileged information to use and how realisable the corrections are. Finally, an ablation on the fine-tuning mechanism in LEAP demonstrates that SFT outperforms DPO and KTO in this setting.

**Strengths:**

- The paper provides a solid theoretical grounding in both the main text and appendix, especially around the analysis on the trade-off between privileged information and the agent's realizability.
- I appreciated the variety and number of domains considered experimentally, which showed the wide applicability of the method.
- The ablations on the fine-tuning methods, across SFT, DPO and KTO were greatly appreciated, as it was otherwise unclear which of these to pick from
- Beyond the general experimental results, I found the questions considered relatively original: whether an agent can bootstrap to self-improvement, whether it can outperform its teacher, and what tradeoffs are faced in terms of amount of privileged information and correction realizability. These were all quite interesting results to read.
- The paper explicitly acknowledges some its limitations, which is commendable and contributes to the paper's transparency and reliability in my view.

**Weaknesses:**

- I found the author's choice of framing decision making as a POMDP somewhat arbitrary and under-justified. There is plenty of valid work that studies decision making using fully observable MDP's, and the connection to the real world is a bit forced
- Aside from the above, the main weakness in my view is that the authors never address/consider/justify the absence of the very obvious alternative: if you are able to train using privileged information anyway, why not simply show this information directly to the student rather than through the teacher? If the privileged information is accessible (by the teacher), this is not really a POMDP anymore really. What analogous real world scenario is this setup trying to address? It appears somewhat ill-defined from what i can tell.
- Somewhat related to the above point, but I found some of the results somewhat obvious and not particularly novel: of course an agent which (indirectly) has additional privileged information will do better than one without such information. It is similarly not surprising that the student eventually outperforms its teacher, given that only the student is trained and is probably comparatively overfitting to the objectively.
- I found section 3.3. rather unclear and perhaps underdeveloped in the main text. For instance, "realizability" on its own is never clearly defined. I would have devoted more of the paper to this analysis, perhaps sacrificing one of the domains from 4.1-4.4 to the appendix if space was necessary.

**Questions:**

My main question is already mentioned above under "weaknesses". If we are allowed to show privileged information to the teacher, which then uses this to correct the student, why not directly show the privileged information to the student, especially given the student-teacher relationship is not adversarial? Which real-world scenario is this setup replicating?

UPDATE 25/11/2024: This question and some of the weaknesses have been addressed in the rebuttal, so I have updated my review score from a 5 to a 6.

---

> ### Author Response · Authors · 2024-11-17
>
> Thank you for your detailed and constructive feedback!
>
> **"Q: I found the author's choice of framing decision making as a POMDP somewhat arbitrary and under-justified..."**
>
> Our choice to frame decision-making as a POMDP is based on the nature of the benchmarks we evaluate on, which are designed around real-world tasks requiring agents to operate with incomplete information. These tasks include locating goal items in web environments (WebShop) or game environments (AlfWorld) and writing code while receiving execution feedback from a bash terminal (InterCode).
>
> In these settings, the agent’s observations are inherently local:
> * **WebShop.** When the agent is on a specific webpage, it only sees the content of that page and has no information about items on other pages.
> * **AlfWorld.** When the agent is at "counter 1," it can only see the objects on the countertop and has no visibility of items at other locations.
> * **InterCode.** When the agent runs a bash command, it observes only the output of that command and lacks knowledge of the underlying directory structure or the state of the Docker container.
>
> This naturally aligns with a POMDP formulation, where the agent’s observations correspond to its local view, and the full state encompasses either the web server, game, or Docker environment.
>
> Moreover, the benchmark papers for AlfWorld, WebShop, and InterCode themselves adopt a POMDP formulation.
>
> **"Q: ... why not simply show this information directly to the student rather than through the teacher? If the privileged information is accessible (by the teacher), this is not really a POMDP anymore really...**
>
> It is crucial to note that at test time, the problem is indeed a POMDP, and the underlying state is not accessible to any method, including LEAP. The student policy must be able to solve the problem from observations alone. Hence an alternate approach that trains the student directly on the privileged state would not be feasible as the state is not available as input at test time.
>
> It is only at train time that we assume that learner may have access to the underlying MDP. Note that these are for train time tasks that are distinct from those at test time. This is made possible by training on a simulator that already maintains this rich state to correctly simulate transitions and evaluate success. We are simply exposing this state to the agent. While the agent can utilize this privileged information during training to improve learning, it cannot rely on it at test time.
>
> This setup reflects real-world scenarios where agents are developed in controlled environments or simulators where richer information may be accessible, but is unavailable in real-world deployment. The goal is to efficiently train the agent to solve the problem using this privileged state, but then distill the policy into a student that works from observations alone. In related works (section 5), we show that such a paradigm has been used extensively in robotics for training robots in simulation and deploying in real world for applications like self-driving and legged locomotion. We build on this paradigm, and propose novel extensions in the form of constrained privileged experts to adapt this for training LLM agents.

---

> ### Author Response · Authors · 2024-11-17
>
> **"Q: ... of course an agent which (indirectly) has additional privileged information will do better than one without such information. It is similarly not surprising ..."**
>
> We emphasize that neither the student nor the teacher has access to any privileged state at test time. The teacher (GPT4) and the student (Llama-8b) get the exact same prompt and observations. The student not only surpasses the teacher, but also surpasses baselines from prior work that use GPT-3.5 and in-context examples showing how to solve tasks.
>
> **On the novelty of results.** We find these results exciting and our qualitative analysis reveals that the student indeed learns to effectively plan and backtrack better than baselines. Given this is the first instance of fine-tuning an LLM agent from corrections on these popular benchmarks, we believe this is an important result for future work to compare against. Finally, as the reviewer already indicated in “Strengths”, our ablations results too are novel and highlight important nuance to the story. For example, we show that the seemingly obvious idea of imitating a teacher with privileged information can actually lead to worse performance due to large realizability gaps.
>
> **On the concern of overfitting.** We first note that train and test tasks are distinct, meaning the privileged states used during training do not transfer to the test tasks. For example, in AlfWorld and Webshop, the target objects and their locations are different in the train and test tasks. Similarly, for InterCode bash, the train and test directory structures are different. As a result, the student agent must learn general strategies to solve tasks effectively at test time.
>
> In AlfWorld specifically, the test tasks are divided into two categories: (1) *In-distribution tasks:* These have similar objectives to the training tasks but with items in different locations. (2) *Out-of-distribution tasks:* These feature entirely new objectives that were unseen during training. We observe significant performance improvements on both categories (Tables 1, 2). This indicates that the student’s success is not due to overfitting to either the privileged states (e.g., item locations) or the task objectives. Instead, the improvements demonstrate the agent’s ability to generalize effectively across tasks and settings.
>
> **"Q: I found section 3.3. rather unclear and perhaps underdeveloped in the main text ... "**
>
> Thanks for the suggestion! We agree that section 3.3 could be improved to be more self-contained. Due to space constraints, we moved the rest of the analysis to Appendix B. We will revise this by bringing those details back into section 3.3. To create space for this, we will move one of the domains from sections 4.1-4.4 to the appendix.

---

> > ### Author Response · Authors · 2024-11-25
> >
> > We hope our response was able to address your concerns. Please let us know if there is anything else we can provide. Thanks!

---

> ### Comment · Reviewer_nN6Y · 2024-11-25
>
> Thank you for the detailed response. I feel like you have addressed my main concerns, particularly by mentioning the application to sim-to-real solutions. I do think the presentation could still use some work, and would benefit from a clearer outlining of the motivations and the downstream use-cases/applications.
>
> Nevertheless I am now convinced of the contribution of this paper and will update my score from a 5 to a 6.
>
> Thanks again and nice work.

---

> > ### Author Response · Authors · 2024-11-26
> >
> > Thank you so much for taking the time to review our work and helping improve it with your feedback. We really appreciate it!
> >
> > We'll be sure to update the draft to better motivate the paper. We'll also emphasize the downstream applications more clearly, including the connection to sim-to-real.

---

### Official Review · Reviewer_E725 · 2024-11-03

**Soundness:** 3
**Presentation:** 2
**Contribution:** 2
**Rating:** 3
**Confidence:** 4

**Summary:**

This is paper is about automatic self-improvement from errors during task execution. They proposed LEAT in which LLM agents as students use feedback from expert teachers to improve itself. The teachers hav access to a privileged state —information available during training. So then even weak teachers are effective. They evaluate that on including text-based games, web navigation, and interactive coding. The results show that it outperforms state-of-the art baselines, enables weak student models (e.g., Llama3-8B) to exceed the performance of strong teacher models (GPT4-o), and (3) allows weak models to self-improve using privileged versions of themselves.

**Strengths:**

- well written.

**Weaknesses:**

- Lack of good reference! Relevant references are missed in introduction! It needs to provide proper reference for each claim!
- The issue of experts is how to prepare knowledge for the experts, how to distinguish between different experts. This is not clear in this work and it needs manual set up which is not applicable.
- They are very similar works in the literature but with different names instead of experts! In that way the innovation of the work is very limited.
- No Fair comparison at all!! The work did not consider fair baselines! There are a lot of up to date methods that works better than ReAct and BUTLER! There are a lot of works from self-correctness that they do the same things.
- In general the contribution of the work is very limited! The teachers should know privilege knowledge however it is not always available! Even with that still prompting and few shot are needed which makes it the same as other prompting works or self correction works but here the authors just used different terms.  And it is very trivial that there should be some improvement vs e.g. react since it is literary using extra information of the environment and more over it uses a suggestion from a larger LLM with that extra information.

**Questions:**

- How to provide those privilege state in more complicated tasks?
- Trade-off between using privileged information and student realizability is not clear!
- Compute privileged state s_t for every datapoint in Di in the algorithm is not clear. That is the limitation part of the algorithm.
- how?  For each time step in Di, LEAP computes the privileged state s_t by leveraging information available only during training
- What is the performance of LEAP vs ReAct but using the same privilege state?

---

> ### Author Response · Authors · 2024-11-20
>
> Thank you for your time and feedback!
>
> **"Q: Lack of good reference! ..."**
>
> We do discuss related work extensively in Section 5. We’ve updated the draft with more references in the Introduction as well. We are happy to include any other references that the reviewer has in mind.
>
> **"Q: The issue of experts is how to prepare knowledge for the experts, ... needs manual set up"**
>
> We would like to clarify that creating an expert for a domain does not require manual setup. We use a nearly identical expert prompt across all three domains we evaluate on (text games, web navigation, and interactive coding).
>
> The primary variation between domains lies in the privileged state provided to the expert. This is represented as a templated {privileged_state} in the prompt, which is automatically extracted for each task ID. This extraction is done either by processing human demonstrations or evaluation metrics or simulator states.
>
> **"Q: They are very similar works in the literature ..."**
>
> We discuss related works extensively in Section 5. Most prior works in this area either do not address interactive environments where agents act and receive feedback from an external environment (e.g., constitutional AI, LEMA) or focus on self-refinement at test time without fine-tuning models (e.g., Reflexion, Self-refine). In contrast, our approach, LEAP, fine-tunes agent models that interact with external environments and enables self-improvement using privileged versions of the agent as a teacher. This distinction is a key contribution of our work, supported by improvements across three diverse benchmarks. If there are specific methods or works we may have missed, we are happy to include them for a more comprehensive comparison.
>
> **"Q: No Fair comparison at all!! ..."**
>
> On ALFWorld, we select contemporary baselines, all from 2023–2024, to ensure relevance and fairness. ReAct, being the closest baseline, is included in the table. To address the reviewer's concern, we have updated the ALFWorld table to include additional clusters of work, even though their approaches and assumptions differ significantly. Specifically, we now include Reflexion and AdaPlanner (which allow multiple tries at test time, unlike LEAP, which tries only once), Autogen (which employs a multi-agent architecture, whereas LEAP uses a single agent), and Expel (which uses retrieval-augmented generation, while LEAP does not rely on retrieval). LEAP outperforms all these methods despite using a much smaller model (Llama-3-8B) compared to GPT-3.5 or GPT-4 used in the other approaches. If there are any other baseline we might have missed, please let us know and we are happy to include them.
>
> **"Q: ... The teachers should know privilege knowledge however it is not always available! Even with that still prompting and few shot are needed which makes it the same as other prompting works .."**
>
> It is important to note that neither LEAP nor ReAct use privileged information during inference, ensuring fair comparisons. Privileged information is exclusively used during training and is restricted to the training set of tasks, it is not applied to any test tasks. Additionally, unlike prompting baselines, LEAP employs an iterative fine-tuning process for model improvement.
>
> **On the availability of privileged information:** We show that privileged information is available across many domains and can often be easily derived from existing sources. In our three evaluation domains, we extract privileged information automatically from pre-existing data or simulators, requiring no additional data collection effort.
>
> In particular, one can think of three common sources of privileged information for any domain:
> * *Simulator States.* Available in most agentic domains requiring interaction with simulated environments before real-world deployment. For instance, in AlfWorld, while agents observe only their current location, the simulator provides full item locations as privileged information during training.
> * *Evaluation Criteria.* Metrics or success criteria are available in nearly every domain. For example, in WebShop we use the product attributes, options, target prices, and in InterCode we use the goal commands and unit tests.
> * *Human Annotations.* While not universally available, they can often be derived from existing human demonstrations. For example, in AlfWorld, we extract key subgoal information (e.g., "drawer 3 contains book 2") from human demonstrations. Similarly, dialog agents often rely on conversation-level subgoal annotations.

---

> > ### Author Response · Authors · 2024-11-20
> >
> > **"Q: the contribution of the work is very limited!"**
> >
> > **On novelty:** We would like to emphasize the novelty of our approach, results, and ablation studies.
> > * To the best of our knowledge, LEAP is the first iterative learning framework for LLM agents that fine-tunes these agents using privileged expert feedback, effectively balancing privileged information and realizability.
> > * Our results show that weak models (e.g., LLama3-8b) can learn to outperform much stronger teachers (e.g., GPT-4o) through successive fine-tuning iterations. Qualitative analysis further shows that the student model learns to plan and backtrack more effectively than baselines. Given this is the first instance of fine-tuning an LLM agent from corrections on these popular benchmarks, we believe this is an important result for future work to compare against.
> > * Our ablation studies introduce novel insights that add depth to the findings. For instance, blindly imitating a teacher with full access to privileged information can in fact lead to worse performance due to significant realizability gaps. Additionally, our ablations reveal that models can also bootstrap and self-correct by leveraging privileged versions of themselves.
> >
> > **"Q: How to provide those privilege state in more complicated tasks? Compute privileged state s_t for every datapoint in Di in the algorithm is not clear..."**
> >
> > We automatically extract privileged states for each task from either existing human demonstrations, evaluation metrics or simulator states. See our response above on “three common sources for privileged information”. For details, please refer to the supplementary code in *scripts/dataproc/extract_privileged_state_*.py* for the exact code. Note that the extraction of state for all tasks, simple or complicated, is automatic and doesn’t require manual work. We will include more details on this in the appendix.
> >
> > **"Q: Trade-off between using privileged information and student realizability is not clear!"**
> >
> > The trade-off is as follows: If the teacher uses 100% privileged information, it is better able to solve the problem and provide optimal actions. However, the teacher’s reasoning may directly reference the privileged state, making it impossible for the student to imitate since the student does not have that information, i.e. the teacher is unrealizable to the student. On the other hand, if the teacher uses 0% privileged information, it may not know how to solve the problem and choose poor actions. However, the teacher’s reasoning is realizable and easy for the student to imitate since they both have the same information.
> >
> > We show qualitative and quantitative examples in Section 4.5 and Figure 5, and state it theoretically in Theorem 3.2. We are happy to expand on any particular aspect that is not clear.
> >
> > **"Q: What is the performance of LEAP vs ReAct but using the same privilege state?"**
> >
> > The performance of LEAP vs. ReAct cannot be evaluated using the same privileged state because neither method uses privileged state at test time. Both rely solely on observations during testing, with the student model in LEAP required to solve the problem independently of any privileged information.

---

> > > ### Author Response · Authors · 2024-11-25
> > >
> > > We hope our response was able to address your concerns. Please let us know if there is anything else we can provide. Thanks!

---

> > ### Comment · Reviewer_E725 · 2024-11-25
> > **Response to Authors**
> >
> > - Lack of performance: I did see the related work!! By lack of reference, as I have written I meant in the introduction. The introduction explained a story. For each part of the story, you should refer to the existing works that already done that.
> > - again the definition of experts is not clear and technical. It is very vague explanation of teacher. How does it confirm that reasoning and actions from teacher is trustable?
> > - This work can be categorized in self-correctness works! They do work with interactive environments and they do use reasoning to improves the next suggestions. These are not limited to the following example but with a very quick search:
> > - Recursive introspection: Teaching language model agents how to self-improve
> > -Selfee: Iterative self-revising llm empowered by self-feedback generation
> >
> > - Baselines: They just compared against ReAct and BUTLER!! These works are very old for this area to compare with. These two works are from 2020 and 2022, very basic possible idea of interactive agent. How is that: "we select contemporary baselines, all from 2023–2024, to ensure relevance and fairness."!
> >
> > - On the availability of privileged information --> three options are just explained which again makes me sure that this is not a technical explanation of the privileged and expert! There is no clear methodology presented; it is more like storytelling than a structured analysis.
> >
> >
> >
> > I firmly stand by my decision to reject this paper .

---

> > > ### Author Response · Authors · 2024-11-30
> > >
> > > * Lack of references: We clarify that we had indeed updated the draft earlier with 11 citations in the introduction. We are happy to add more if necessary, provided the reference and the appropriate sentence that requires it.
> > >
> > > * Definition of experts is not clear and technical: We devote an entire subsection 3.1 of the paper to rigorously define a teacher as a policy $(\tilde{\rho}_t, \tilde{a}_t) \sim \pi^E(\cdot | {\rho}_t, {a}_t, h_t, s_t) $. We formally define the POMDP in section 2. In the analysis section in Appendix B, we carefully define the performance of the expert $J(\pi^E)$, and prove theoretical bounds on LEAP with respect to the performance of the expert. We argue that the expert is clearly defined in math and are happy to clarify any errors in specific equations.
> > >
> > > * Self-correction references: We first note that the focus of LEAP is not on self-correction, but rather using privileged information to provide on-policy corrections. We ran self-correction as an ablation to show the flexibility of LEAP as a framework. Our related work already covers self-correction papers like Self-Correct (Welleck et al., 2023), which serve as a foundation for later works such as RISE (Qu et al., 2024), as acknowledged by the RISE authors themselves. We have included RISE to the updated draft, but note that there are several key distinctions: (1) RISE does not involve external environments and tests on math reasoning datasets such as GSM8K. In contrast, LEAP operates with external environments, testing on popular agent benchmarks. External environments introduce unknown dynamics in the agent’s trajectories, making the problem harder. (2) RISE requires solving a reinforcement learning problem that is sample inefficient. In comparison, LEAP leverages privileged information and solves this as an imitation learning problem. The reviewer also mentioned Selfie, which we note is a blog about self-refining LLMs. We already include references to papers like Self-refine (Madaan et al., 2023) where LLMs refine their responses based on feedback from their own outputs.
> > >
> > > * Baselines: We clarify that we had indeed updated the draft earlier with 4 new baselines (see Table 1): Reflexion, Adaplanner, Autogen, Expel, which are all from 2023-2024. LEAP outperforms all of these baselines, despite some of them having multiple attempts at the MDP compared to a single attempt for LEAP.
> > >
> > > * “There is no clear methodology presented; it is more like storytelling than a structured analysis.“ : We strongly oppose this statement given the rigorous analysis we present in appendix B and section 3.
> > >     * We clearly define the problem as a POMDP where the agent receives observations $o_t$ and define the privileged state $s_t$ as the state of the underlying MDP.
> > >     * The privileged expert $\pi^E(s_t)$ observes the privileged state, while the student must operate on history of observations and action $h_t$.
> > >     * Definition 3.1 (also Def B.2) defines the average realizability gap $\epsilon (\pi^E, T)  := \sup_{\pi} \frac{1}{T} \sum_{t=1}^{T} \mathbb{E}_{s_t, h_t \sim d^\pi_t} || \pi^E(a_t|s_t) - \pi^{\star}(a_t | h_t)) ||_1$ between the expert and the best student policy
> > >     * Theorem 3.2 (also Thm B.4) provably bounds the performance of the LEAP policy $J(\pi)$ in terms of the performance of the privileged expert  $J(\pi^E)$ and the realizability gap. We do this by leveraging the performance difference lemma, and no-regret online learning.
> > >     * Definition B.5 then introduces a constrained privileged expert
> > > $    \pi^E_{\delta}(.|s_t, h_t) := \arg\min_{\tilde{\pi}} \mathrm{KL}\left( \tilde{\pi}(.|s_t, h_t) || \pi^E(.|s_t) \right) \quad \mathrm{s.t.} \quad \mathrm{KL}\left( \tilde{\pi}(.|s_t, h_t) || \pi^E(.|h_t) \right) \leq \delta$ that minimizes the KL divergence to the privileged expert while staying within a $\delta$ KL ball of an expert that does not have privileged state.
> > >     * Theorem B.6  provably bounds the performance of the LEAP policy $J(\pi)$ in terms of the performance of the constrained privileged expert and the realizability gap wr.t. it. By increasing $\delta$, we formally trade off between leveraging privileged information vs decreasing realizability.
> > >     * We empirically show that this tradeoff exists in the experiment in Section 4.5

---

### Official Review · Reviewer_EhvG · 2024-11-04

**Soundness:** 3
**Presentation:** 2
**Contribution:** 3
**Rating:** 6
**Confidence:** 3

**Summary:**

The paper proposes LEAP, an iterative fine-tuning framework that continually improves LLM agents using feedback from AI expert teachers. The key insight is to equip the expert teachers with a privileged state - additional information available during training but hidden at test time. This allows weak experts to provide guidance, improving the student agent's performance. LEAP is evaluated on diverse decision-making benchmarks, including text-based games, web navigation, and interactive coding, and shows that it outperforms state-of-the-art baselines, enables weak student models to exceed the performance of strong teacher models, and allows weak models to self-improve using privileged versions of themselves.

**Strengths:**

* The proposed method looks valid and outperforms the baseline (ReAct).
* Evaluate three diverse environments: ALFWorld, WebShop, and Intercode.
* Provide detailed experiment and theoretical analysis of the trade-off between realizability and privileged information.
* The proof appears to be correct, and the prompts are included in the Appendix.

**Weaknesses:**

* The analysis of the costs associated with iterative fine-tuning and calling LLM experts was not included. While the authors assert that their method is not comparable to Reflexion [1], Reflexion appears to be a more cost-effective approach for enhancing the performance of LLM agents with minimal trials.
* It seems somewhat inequitable to compare this method with those that do not utilize privileged information, as such information is typically not available in real-world scenarios.

**Questions:**

**Q1:** In the proof of theorem B4, there is a step using the performance difference lemma, but I did not see the connection here. Could you provide the derivation of how you get the inequality?

**Q2:** In the contribution section, Webshop is reported as achieving 61.8%. However, the metric used in the experiment is the score, which is usually much higher than the success rate. Could you clarify what metrics you are actually evaluating?

**Q3:** Could you provide the self-correct results (using Llama3-8B as expert teacher) for WebShop and Intercode?

The overall quality of the paper is good, but there are still some weaknesses and questions that need to be addressed. I will reassess the score based on the authors' clarifications.

---

> ### Author Response · Authors · 2024-11-16
>
> Thank you for your thoughtful evaluation and insightful feedback!
>
>  **"Q: ... Reflexion appears to be a more cost-effective approach for enhancing the performance of LLM agents with minimal trials."**
>
> Approaches like Reflexion are test-time refinement methods that do not involve any training. The process is as follows: At test time, the agent starts with an empty memory and generates a trajectory by interacting with the environment. If the trajectory fails, it prompts a self-reflection module to summarize the failure and add to the memory. The agent then retries with this updated memory, iterating until success or a max retry limit.
>
> This impacts both cost-effectiveness and generalizability,
>
> * **Inference Costs.** While Reflexion avoids training costs, it incurs significantly higher inference-time costs due to multiple retries per task. For example, on AlfWorld, Reflexion requires ~6 retries to match LEAP's performance. Over 30 time steps, this results in 180 actions (30 * 6) compared to LEAP's 30 actions (30 * 1) during inference.
> * **Test-time Learning.**  Reflexion improves through repeated attempts on test tasks, effectively refining directly on the test set. In contrast, LEAP trains on failures from a separate training set, allowing it to better generalize and make only a single attempt at the test task.
> * **Cross-task learning.** Reflexion’s learning is confined to task-specific memory at test time, preventing knowledge transfer across tasks. As a result, it lacks the ability to generalize from failures encountered in different tasks.
>
> **"Q: ... somewhat inequitable to compare this method with those that do not utilize privileged information, as such information is typically not available in real-world scenarios."**
>
> We would like to clarify that none of the methods, including LEAP, use privileged information during inference, ensuring a fair comparison. In LEAP, privileged information is strictly limited to the training phase and restricted to the training set tasks.
>
> Although privileged information may appear unavailable in real-world scenarios, it is often accessible and can be derived from existing sources across many domains. In our three evaluation domains, we extract privileged information from pre-existing data, requiring no additional data collection effort.
>
> In particular, one can think of three common sources of privileged information for any domain:
> * **Simulator States.** Available in most agentic domains requiring interaction with simulated environments before real-world deployment. For instance, in AlfWorld, while agents observe only their current location, the simulator provides full item locations as privileged information during training.
> * **Evaluation Criteria.** Metrics or success criteria are available in nearly every domain. For example, in WebShop we use the product attributes, options, target prices, and in InterCode we use the goal commands and unit tests.
> * **Human Annotations.** While not universally available, they can often be derived from existing human demonstrations. For example, in AlfWorld, we extract key subgoal information (e.g., "drawer 3 contains book 2") from human demonstrations. Similarly, dialog agents often rely on conversation-level subgoal annotations. In fact, subgoal annotations are significantly easier to collect compared to dense demonstrations, making them a practical alternative as long as approaches (like LEAP) can use such information.

---

> ### Author Response · Authors · 2024-11-16
>
> **"Q1: ... Could you provide the derivation of how you get the inequality?"**
>
> We have added a derivation for using the PDL to get to the inequality and updated the draft on OpenReview. Please refer to Theorem B.4, Eqs (10) - (12).
>
> **"Q2: ... Could you clarify what metrics you are actually evaluating?"**
>
> Thanks for pointing this out. For WebShop, the metric we are evaluating is indeed the score, which is a weighted combination of individual metrics such as product attribute, type, option, and price. For AlfWorld and InterCode, the metric used is the success rate, defined as the percentage of trajectories that result in a final success.
>
> We have fixed the contribution section and added this as clarification in the WebShop setup: “Performance is evaluated using an overall score with $4$ components on whether: attributes match (\textsc{Att}), options match (\textsc{Opt}), product types match (\textsc{Type}) and price matches (\textsc{Price}). Note that the score is different from success rate, and is typically higher as it assigns partial credit for matching 1 or more components.”
>
> **"Q3: Could you provide the self-correct results (using Llama3-8B as expert teacher) for WebShop and InterCode?"**
>
> The self-correct / self-improve experiments we conducted were intended as an additional ablation. Our hypothesis is that in domains where we have richer privileged information (e.g. AlfWorld), the privileged information itself plays a much stronger role in driving performance improvements compared to the size of the teacher model. As a result, even with weaker teachers, as in self-improvement settings, the corrections tend to be of high quality.
>
> We are happy to extend this ablation to other domains as well. Due to time constraints, we can likely complete the ablation for one additional domain now and include the results for the third domain in the final version. We will report the results back here as soon as they are available.

---

> > ### Author Response · Authors · 2024-11-22
> >
> > **Self-correct results for Webshop**
> >
> > As promised, we ran the ablation of self-correction for Webshop and present the results below:
> >
> > | Model   | Score | Num actions | r_att | r_option | r_type | r_price |
> > |---------|-------|-------------|-------|----------|--------|---------|
> > | iter 0  | 29.4  | 21.1        | 28    | 25.8     | 35     | 36.8    |
> > | iter 1  | 36.8  | 18.7        | 36.6  | 33       | 44.2   | 45.2    |
> > | iter 2  | 39.8  | 18.4        | 39.9  | 31.7     | 48.1   | 51      |
> >
> > We make the following observations
> > * We see self-correction improve consistently over iterations (29.4 -> 36.8 -> 39.8)
> > * The performance improvements are lower with Llama-8B teacher vs GPT-4o teacher. This is in line with our hypothesis that for environments like Webshop, where the privileged information is not as rich as ALFworld, the performance improvements do depend on the strength of the teacher to provide meaningful corrections.
> >
> > We will add these results along with self-correction on Intercode to the final draft of the paper.

---

> > > ### Author Response · Authors · 2024-11-25
> > >
> > > We hope our response was able to address your concerns. Please let us know if there is anything else we can provide. Thanks!

---

> ### Comment · Reviewer_EhvG · 2024-11-25
>
> Thank you for the clarifications and additional experiments. Most of my concerns have been addressed; however, I still have two points I’d like to raise:
>
> 1. While I understand that privileged information is only used during training and recognize the difficulty in directly comparing methods that also leverage such information, given that this is a central contribution of your work, I still feel that the significant performance improvement may largely stem from the stronger information utilized. This makes the baseline comparison less effective.
>
> 2. In the additional experiment on Webshop using self-correction, the results appear lower than ReAct, which highlights the reliance on a strong teacher as a potential limitation.
>
> Overall, I appreciate the analysis of the trade-offs, along with the theoretical proof, and I believe this work has the potential to inspire further exploration in this area. I would maintain my rating of 6.

---

> ### Author Response · Authors · 2024-11-26
>
> Thank you so much for taking the time to review our work and helping improve it with your feedback. We really appreciate it!
>
> 1. While we agree that privileged information plays an important role, test time improvements stem largely from *general* strategies that the student model was able to learn rather than overfit to the privileged information itself. We note this happens due to 2 key components of LEAP: (1) on-policy feedback and (2) teacher giving realizable corrections without revealing privileged information.
>
>     For example, in AlfWorld (Fig. 2), the student is able to learn general strategies for efficient exploration. At training, the student learns that instead of sequentially opening drawers, it should instead prioritize likely locations for items, e.g., "Cellphones are commonly found on desks". Such a strategy transfers over to test tasks with different objectives, e.g. “I should check the dresser, as it often contains valuable items”.
>
>
> 2. Our hypothesis is that if the privileged information is strong enough, even a weak teacher enables significant improvements. While a stronger teacher always helps, its impact is less critical when privileged information is sufficiently rich.
>
>     For example, in AlfWorld, the privileged state contains more information allowing a 8B model to bootstrap itself from 65.7% → 82.1% → 91.8%, outperforming a ReAct GPT-4o model (54.3%). When using GPT-4o as the teacher, the maximum performance improves further to 96.4%, demonstrating that while a strong teacher provides an additional boost, privileged information plays a more substantial role in driving performance. In WebShop, the privileged information is comparatively weaker, resulting in a larger performance gap between the model using itself as the teacher and the model using a stronger GPT-4o as the teacher.

---

> > ### Author Response · Authors · 2024-11-30
> >
> > To further address your concern that LEAP’s performance gain might primarily stem from the use of privileged information, we constructed a new baseline, SFT-privileged. This baseline collects demonstrations from a privileged teacher that utilizes the same privileged information as LEAP and trains a student policy using Supervised Fine-Tuning (SFT). This approach is referred to as context distillation [1], where privileged information is added to the teacher's context and distilled into the student policy. To ensure a fair comparison, we used the same prompt structure as LEAP, ensuring the teacher does not explicitly reveal privileged information to the student.
> > We run this baseline on ALFWorld and present the results below:
> > | Model                        | Success Rate (%) | Num actions | Num actions when success |
> > |------------------------------|------------------|-------------|------------------------|
> > | SFT-privileged | 42.5            | 20.8       | 8.4                    |
> > | LEAP iter 0                 | 65.7            | 18.6       | 12.7                   |
> > | LEAP iter 1                 | 91.0            | 11.9       | 10.1                   |
> >
> > **Key observations:**
> > * SFT-privileged has a significantly lower success rate (42.5%) than both LEAP iterations (Iter 0: 65.7%, Iter 1: 91.0%).
> > * SFT-privileged requires fewer actions when successful (8.4) compared to LEAP (Iter 0: 12.7, Iter 1: 10.1).
> >
> > **Explanation:**
> >
> > At training time, demonstrations from the privileged teacher are always successful. The teacher provides general reasoning (e.g., “Cellphones are commonly found on desks”) and directly utilizes privileged information to locate objects and solve tasks. The SFT-privileged student is trained exclusively on these successful demonstrations.
> > At test time:
> > * **When correct,** the SFT-privileged student imitates the reasoning and succeeds with an optimal number of actions. This explains why the number of actions when successful (8.4) is lower than LEAP's.
> > * **When incorrect,** the student struggles to recover from errors. Having been trained only on successful demonstrations, it lacks the ability to handle failure cases. For example, it might hallucinate incorrect objects or repeatedly visit the same location. This inability to recover leads to a significantly lower success rate (42.5%).
> >
> > **Summary:**
> >
> > The improved performance of LEAP is driven not just by privileged information but by two key components:
> > 1. **On-policy corrections:** On-policy corrections enable the student to recover from errors, as demonstrated by the significant improvement over the SFT-privileged baseline (91.0% vs. 42.5%).
> > 2. **Realizable corrections:** Realizable corrections, where the teacher avoids revealing privileged information, also play a critical role in LEAP’s performance. Ablations show a substantial drop when realizable corrections are removed (91.0% vs 5.2%).
> >
> > Thank you for your feedback, which allowed us to highlight the importance of these components more effectively.
> >
> > **Reference**
> > [1] Snell, C., Klein, D., & Zhong, R. (2022). Learning by distilling context. arXiv preprint arXiv:2209.15189.

---

> > > ### Comment · Reviewer_EhvG · 2024-12-03
> > >
> > > I appreciate the author for including the additional baseline. While naive imitation learning is unlikely to achieve even normal performance, at least it is still a baseline leveraging privileged information. I maintain my positive score.

---

### Meta-Review · Area_Chair_wgVi · 2024-12-23

**Metareview:**

This paper proposes an interesting framework to improve LLM agents by utilizing a LLM teacher with access to privileged information (the ground truth states that may be hidden in practice) to correct the reasoning and action trace of the student model, and iteratively fine-tune on such corrected traces. This iterative process can obtain student models outperforming the teacher model. While acquiring privileged information can be hard for certain applications like search agent in practice, the idea and observations in this paper looks promising and can be applied once the privileged information can be obtained.

**Additional Comments On Reviewer Discussion:**

The review provided by reviewer E725 is not clear and looks biased, therefore I have ignored the review. Main concerns of other reviewers were around the availability of the privileged information in practice, but there can be ways to obtain such oracle information if one decides to invest in this approach (e.g., through human data collection). The observation that iteratively correcting the student's reasoning trace can obtain students outperforming the teacher looks quite promising.

---

### Decision · Program_Chairs · 2025-01-22

Accept (Poster)